# Profiling of hMPV F-specific antibodies isolated from human memory B cells

Xiao Xiao[1,2,3], Arthur Fridman[4], Lu Zhang[5], Pavlo Pristatsky[6], Eberhard Durr [1], Michael Minnier[7], Aimin Tang[1], Kara S. Cox[1], Zhiyun Wen[1], Renee Moore[2], Dongrui Tian[8], Jennifer D. Galli[1], Scott Cosmi[9], Michael J. Eddins[10], Nicole L. Sullivan[1], Xiaodong Yan[8], Andrew J. Bett[1], Hua-Poo Su[10], Kalpit A. Vora [1,11✉], Zhifeng Chen [1,11✉] & Lan Zhang [1,11✉]

Human metapneumovirus (hMPV) belongs to the *Pneumoviridae* family and is closely related to respiratory syncytial virus (RSV). The surface fusion (F) glycoprotein mediates viral fusion and is the primary target of neutralizing antibodies against hMPV. Here we report 113 hMPV-F specific monoclonal antibodies (mAbs) isolated from memory B cells of human donors. We characterize the antibodies' germline usage, epitopes, neutralization potencies, and binding specificities. We find that unlike RSV-F specific mAbs, antibody responses to hMPV F are less dominant against the apex of the antigen, and the majority of the potent neutralizing mAbs recognize epitopes on the side of hMPV F. Furthermore, neutralizing epitopes that differ from previously defined antigenic sites on RSV F are identified, and multiple binding modes of site V and II mAbs are discovered. Interestingly, mAbs that bind preferentially to the unprocessed prefusion F show poor neutralization potency. These results elucidate the immune recognition of hMPV infection and provide novel insights for future hMPV antibody and vaccine development.

[1] Infectious Diseases and Vaccines Discovery, Merck & Co., Inc., West Point, PA, USA. [2] Discovery Biologics, Merck & Co., Inc., Boston, MA, USA. [3] MRL Postdoctoral Research Program; Merck & Co., Inc., Kenilworth, NJ, USA. [4] Data Science and Scientific Informatics, Merck & Co., Inc., Rahway, NJ, USA. [5] Bioinformatics and Biomarker Research, MSD, Beijing, China. [6] Analytical Research and Development, Merck & Co., Inc., West Point, PA, USA. [7] AgileOne, Torrence, CA, USA. [8] Wuxi Biortus Biosciences Co. Ltd., Wuxi, China. [9] Eurofins PSS Insourcing Solutions, Lancaster, PA, USA. [10] Computational and Structural Chemistry, Merck & Co., Inc., West Point, PA, USA. [11] These authors contributed equally: Kalpit A. Vora, Zhifeng Chen, Lan Zhang. ✉email: kalpit.vora@merck.com; zhifeng.chen@merck.com; lan_zhang2@merck.com

Pneumoviruses including respiratory syncytial virus (RSV) and human metapneumovirus (hMPV) are common causes of acute lower respiratory infection[1–6]. Like RSV, hMPV infection leads to a variety of symptoms including coughing, wheezing, pneumonia, and bronchiolitis. High-risk populations, including infants, young children, elderly people, and immuno-compromised patients, are more likely to develop severe symptoms requiring hospital management[7–14]. Co-infection with other respiratory pathogens is common. Recently, cases of co-infection of SARS-CoV-2 and hMPV have been reported[15–17], and such co-infection with hMPV appeared to alter the function of infected monocytes and dampen interferon response in a severe COVID-19 patient[15]. Despite these medical burdens, currently, there is no approved hMPV vaccine or neutralizing antibody available for therapeutic or prophylactic purposes.

The surface glycoprotein F of hMPV mediates the fusion of viral and cellular membranes[18,19]. The F protein is highly conserved in sequence between different hMPV subtypes, and it appears to be more critical for virus infection than other surface proteins G and SH[20,21]. It has also been reported that F is the primary target of neutralizing and protective antibodies against hMPV infection, whereas antibodies elicited by G and SH proteins are not protective[22–24]. Therefore, F protein is an attractive target for neutralizing antibodies and vaccine development against hMPV infection.

hMPV F is a class I viral fusion protein present as a homo-trimer on the viral surface as well as the membrane of host cells. The precursor of F ($F_0$) is synthesized as an intact polypeptide and is subsequently subjected to proteolytic processing to become functional[19]. Unlike RSV which has two furin-cleavage sites that can be cleaved intracellularly during transport to the cell membrane[25], hMPV only has one protease cleavage site being cleaved at the cell surface or in the virus particle by transmembrane proteases such as TMPRSS2[26]. Cleavage of both RSV and hMPV F generates two disulfide-linked chains ($F_2$ and $F_1$). The fusion peptide, which is located at the N-terminus of $F_1$ right after the protease cleavage site, is buried inside a hydrophobic cavity and interacts with adjacent monomers, presumably stabilizing the trimer conformation[27,28]. This globular "prefusion" (PreF) form of proteolytically processed F trimer is metastable and can be rearranged into a more stable rod-shaped "postfusion" (PostF) form through a series of conformational changes, with the N-terminal fusion peptides inserted into the target cell membrane and mediating virus-cell fusion[19]. Protein engineering efforts have been made to stabilize hMPV and RSV F in PreF and PostF conformations by introducing additional mutations or modifying protease cleavage sites[27–32]. Despite having only about 35% sequence identity, the structures of both stabilized PreF and PostF proteins between RSV F and hMPV F are remarkably conserved[27–31,33,34].

Over the past years, a large number of monoclonal antibodies (mAbs) targeting RSV F antigens have been isolated from human B cell repertoire and characterized by various approaches[33,35–47]. According to these results, six major antigenic sites on RSV F have been defined (Ø, I, II, III, IV, V). PreF-specific antibodies targeting sites Ø and V are immunodominant and account for a large proportion of neutralizing activity in human sera and antigen-specific memory B cells of adults[36,48,49], suggesting that PreF is a better vaccine candidate for RSV to elicit higher neutralizing antibody (nAb) responses than PostF. In contrast, the immunogenicity potential of stabilized hMPV PreF and PostF appears to be comparable, likely due to the large glycan shield that is only observed at the site Ø of hMPV F protein[28]. Although a limited number of hMPV F-specific mAbs have been discovered from mice immunization[50], phage display[51], and human B cells[41,45,47,52–55], a comprehensive understanding of human antibody recognition to hMPV F during natural infection remains

elusive and the antigenic epitopes on hMPV PreF and PostF structures need further characterization.

In this study, we report a large panel of hMPV F-specific mAbs isolated from memory B cells of multiple healthy human donors. Some of the isolated mAbs exhibit ultra-high neutralization potency in vitro. We characterized one of the most potent mAbs that recognizes a PreF-specific epitope on site V. Further in-depth characterization of isolated mAbs reveals diverse antigenic sites on hMPV F, including overlapping but distinct epitopes on pre-defined site II and V, and four antigenic sites that have not been reported previously in RSV F. Unlike RSV, the antibody responses to hMPV F appear to be less dominant against the apex of the antigen. Furthermore, a panel of mAbs that bind preferentially to the uncleaved PreF have been identified, suggesting potential immunogenic differences between unprocessed and processed F antigens. The reported antibodies here and characterization of their antigenic epitopes on hMPV F provide a valuable framework to further understand the structural basis of an immune response against hMPV and guidance for future hMPV antibody and vaccine development.

## Results

**Isolation of a large panel of hMPV F-specific mAbs from human memory B cell repertoire.** To comprehensively profile the human antibody recognition sites to hMPV F antigen, we isolated more than 100 mAbs from human memory B cell repertoires of multiple adult donors by antigen-specific single memory B cell sorting method (Supplementary Fig. 1). A wild-type (WT) hMPV F ectodomain trimer, which showed no obvious cleavage between F1 and F2 (Supplementary Fig. 2) and likely represents a majority of PreF conformation[34], was used as the primary bait for memory B cell sorting. In some cases, a monomeric form of WT hMPV F antigen or a mixture of WT hMPV trimer described above and a prefusion-stabilized RSV F antigen DS-Cav1 were used (Supplementary Data 1). As a result, a total of 113 mAbs from seven donors, including 88 mAbs from three major donors, have been successfully cloned, expressed, and confirmed to bind to hMPV F antigens (Supplementary Data 1). Three recently reported RSV/hMPV cross-neutralizing mAbs (M1C7, M2B6, and M2D2) were also included in this study[56]. Clonal lineage analysis showed that the hMPV F-specific B cell repertoires were highly diverse, with 111 out of 113 antibodies being unique. The germline distribution was also diverse (Fig. 1a), indicating that clonal expansion is rare in hMPV F induced antibody response. The lengths of HCDR3 ranged from 9–24 (median 15) (Fig. 1b and Supplementary Fig. 3A), which is similar to the recently reported RSV F-specific repertoires in adults[36]. The average levels of somatic hypermutation varied between the donors' repertoires (Supplementary Fig. 3B), ranging from 0–47 (median 21) nucleotide substitutions per VH gene and 0–35 (median 11) nucleotide substitutions per VK/VL gene (excluding CDRH3), respectively (Fig. 1c). Eighty-five out of 113 mAbs neutralized both hMPV A or B subtypes in plaque reduction neutralization assays under tested antibody concentrations, 12 showed high neutralizing potency on both hMPV subtypes A and B with IC50 under 50 ng/mL (Table 1). Six mAbs only showed neutralizing activity for hMPV subtype B (Supplementary Data 1, highlighted in orange), likely because F antigens derived from an hMPV subtype B sequence were used for sorting. Thirteen hMPV neutralizing mAbs showed cross-binding to RSV F antigens and cross-neutralization to RSV A and RSV B strains (Supplementary Data 1, highlighted in green).

**Characterization of an ultra-potent PreF-specific hMPV F neutralizing mAb M4B06.** Among the isolated mAbs, M4B06 showed the highest hMPV neutralization potency on both

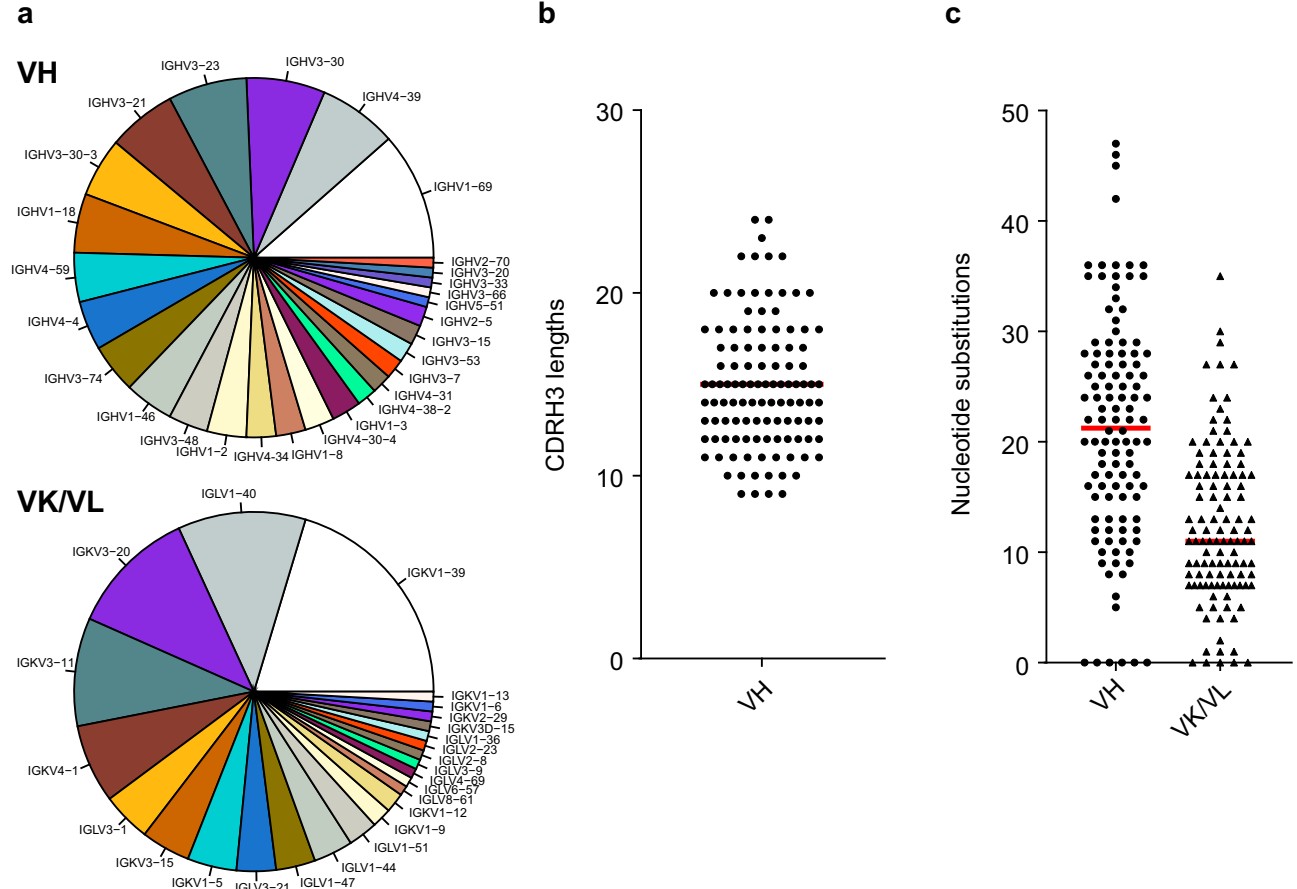

**Fig. 1 CDRH3 lengths and somatic hypermutations (SHM) of hMPV F-specific mAbs. a** Germline frequency of VH and VK/VL. **b** CDRH3 lengths.
**c** Nucleotide substitutions of VH (excluding CDRH3), and VK/VL. Red bars indicate the median. The analysis was based on the Kabat delineation system[75].
Source data are provided as a Source Data file.

tested hMPV A and B subtypes with IC50 of 3.5 and 2.2 ng/mL, respectively (Fig. 2a). Biolayer interferometry (BLI) binding assay showed that M4B06 only bound WT and stabilized PreF protein but did not bind PostF protein, suggesting that it targets a PreF-specific epitope (Fig. 2b). To further understand the epitope of M4B06, we first generated monoclonal antibody-resistant mutants (MARMs) of an hMPV A strain under the selection pressure of M4B06. After three rounds of culture with an increasing concentration of M4B06 Fab, the hMPV virus was able to replicate under the Fab concentration of 40 μg/mL, which is about 500-fold higher than the IC50 against the wild-type virus. Sequencing of single purified plaques of MARMs revealed three different MARMs: V231A/G264R, G264R/Q434H, and K138Q. The neutralization potency of M4B06 IgG to these three MARMs was reduced by 52-, 70-, and 49-fold, respectively, and M4B06 Fab failed to neutralize any of the three MARMs at all, confirming that the mutations generated in these MARMs impaired the neutralizing activity of M4B06 (Fig. 2c). Additionally, we generated hMPV F variants carrying single and double MARM-derived mutations to investigate whether they interrupt M4B06 binding. ELISA binding assay showed that these mutations did not impact the binding to a control antibody M2D2, which targets a different epitope[56], suggesting that the MARM mutants were folded similarly to the WT protein (Supplementary Fig. 4). However, K138Q reduced binding by at least 100-fold and single or double mutants carrying G264R completely abolished M4B06 binding, suggesting that K138 and G264 are critical for M4B06 binding and likely located within the M4B06 epitope (Fig. 2d). K138 and G264 are both solvent-exposed and in proximity to each other,

located at a region equivalent to antigenic site V previously defined in RSV F (Fig. 2e). In a parallel epitope mapping study by hydrogen-deuterium exchange mass spectrometry (HDX-MS), peptides 131–146, 137–146, and 260–266 showed a significantly reduced exchange rate between the hMPV F-M4B06 complex and hMPV F alone, suggesting that these peptides contain the epitope residues of M4B06, which is consistent with the results from the MARM study (Fig. 2e and Supplementary Fig. 5). Moreover, the identified epitope of M4B06 was further confirmed by a Cryo-EM map of the M4B06-hMPV F complex at a resolution of 3.75 Å, which clearly showed that three M4B06 Fabs sit on the epitope area at a nearly perpendicular angle (Fig. 2f and Supplementary Fig. 6). Taken together, we identified a highly potent neutralizing mAb M4B06 targeting an hMPV PreF-specific epitope that is similar to the previously defined site V of RSV F.

**Epitope mapping of hMPV F-specific mAbs.** To map the targeted antigenic sites of all isolated hMPV F-specific mAbs, we first designed and tested a panel of hMPV F epitope mutant proteins based on (1) reported RSV and hMPV F epitopes and mutants; (2) residues that are different between hMPV A and B viruses for antibodies that preferentially target one of the two subtypes; and (3) epitope residues determined by other approaches (See Methods for details). As a result, we have identified a small panel of mAbs showing reduced binding to a subset of PreF mutants (Fig. 3 and Supplementary Fig. 7A) that mapped to the antigenic sites previously defined as site II, site V, and site IV based on studies of RSV F. Specifically, M1D2 and M1C7s were

**Table 1 Characteristics of isolated mAbs with neutralizing IC50 more potent than 50 ng/mL on both hMPV subtype A and B.**

| mAb name | Sites | Neutralization IC$_{50}$ (ng/mL) | | VH | | | | VL | | | |
|---|---|---|---|---|---|---|---|---|---|---|---|
| | | hMPV A | hMPV B | CDR3 sequence | Germline | CDR lengths | Nucleotide substitutions | CDR3 sequence | Germline | CDR lengths | Nucleotide substitutions |
| M1C7s | II | 15 | 9.2 | ATVLDWSNALDI | IGHV1-69*09 | 8,8,12 | 34 | MQGLHLPWT | IGKV2-29*03 | 11,3,9 | 19 |
| M1D2 | II | 11.7 | 12.4 | ARRPPLEYFYYFMDV | IGHV4-4*02 | 9,7,18 | 27 | QQSYSNPYS | IGKV1-39*01 | 6,3,9 | 17 |
| M2A09 | II | 13.3 | 45.6 | ATEGLESGYPPYLEH | IGHV1-69*04 | 8,8,15 | 35 | QQYGASPPT | IGKV3-20*01 | 6,3,9 | 20 |
| M6EI0 | II | 23.7 | 10.4 | ARDRTVIKYADGMDV | IGHV1-69*01 | 8,8,15 | 30 | QQSHTTPLT | IGKV1-39*01 | 6,3,9 | 27 |
| M7D5 | II | 35.4 | 35.9 | AQDRTVTKYAHGLDV | IGHV1-69*01 | 8,8,15 | 27 | QQSHTTPLT | IGKV1-39*01 | 6,3,9 | 30 |
| M2B6 | III | 44 | 23.7 | ASLTITPGWFDS | IGHV4-59*01 | 8,7,12 | 35 | QSYERNLDGAL | IGLV1-40*01 | 9,3,11 | 27 |
| M2B06 | III' | 23 | 20.6 | ARDDQIVVMPAGFDR | IGHV4-4*02 | 9,7,15 | 19 | QQSKSIPYT | IGKV1-39*01 | 6,3,9 | 17 |
| M3C05 | IV | 18.4 | 7.3 | AREFGVLDYHYGMDV | IGHV3-74*01 | 8,8,15 | 13 | ATWDDSLNGPV | IGLV1-44*01 | 8,3,11 | 8 |
| M3C12 | IV | 7 | 1 | ARELGILDYYYGMDV | IGHV3-74*01 | 8,8,15 | 15 | AAWDDSLNGPV | IGLV1-44*01 | 8,3,11 | 9 |
| M3D12 | IV | 23.1 | 14.5 | ARLAVAGVTGFHY | IGHV4-30-4*01 | 10,7,13 | 22 | QQRSNWGLT | IGKV3-11*01 | 6,3,9 | 9 |
| M4B06 | V | 3.5 | 2.2 | AKVPSFFVFEDGFDM | IGHV3-23*01 | 8,8,15 | 26 | AAWDDTLNGYV | IGLV1-44*01 | 8,3,11 | 29 |
| M1C7 | V | 8.3 | 8.5 | VKSGSFSSAFWLED | IGHV3-30-3*01 | 8,8,14 | 28 | QQRSTWT | IGKV3-11*01 | 6,3,7 | 13 |

mapped to site II as they showed reduced binding to hMPV F mutants A238R and R156A/S232A/A238R (Fig. 3). In particular, R156A/S232A/A238R abolished the binding of M1C7s. Multiple binding modes of site V mAbs have been observed: A138 and G264 are critical for M4B06 binding; L130S/G152K/N153A mutant abolished the M3D04 binding; both T143K and N135T/G139N/Y233N disrupted binding of M3C06 whereas only N135T/G139N/Y233N but not T143K affected binding of M4A10 (Fig. 3). In addition, M1A04 has been mapped to site IV as it interacts with residue D414 (Supplementary Fig. 7A).

However, there is a limitation with this approach, and we were not able to map the epitopes of many identified antibodies with these hMPV F epitope mutant proteins. First of all, many designed mutant F proteins exhibited a low expression level and could not be used in our study. Additionally, antibodies can interact with multiple residues on F protein and binding might not be affected by mutation of a single or a few residues. To further characterize all antibodies, we performed BLI-based epitope binning with a panel of reference mAbs representing different epitopes, including the aforementioned M1D2 (site II), M1C7s (site II), M1A04 (site IV), M4B06 (site V), previously reported mAb DS7 (DS7-site)[34], and three RSV/hMPV cross-neutralizing mAbs M2B6 (site III), M2D2 (site IV), and M1C7 (site V)[56]. Based on the preliminary binning results, M4B11, M2A05, M8C10, and M1A08 were included as additional reference mAbs for epitope binning to characterize those mAbs that did not show significant competition with the first group of reference mAbs described above (Fig. 4a).

Based on epitope binning, hMPV F-specific mAbs were categorized into eight groups (Fig. 4a). Four groups of mAbs showed typical competition profiles similar to mAbs targeting previously defined sites V, II, III, IV on RSV F, thus these mAbs were mapped to sites V, II, III, IV, respectively. The other four groups of mAbs showed unique competition profiles that have not been previously reported for RSV F: (1) The first group of mAbs significantly competed with site III mAb M2B6 and hMPV mAb DS7, and showed less competition with site II mAbs M1C7s and M1D2, suggesting that they targeted a region between classical site III and DS7-site. Unlike classical site III mAbs that cross-neutralize both hMPV and RSV, this group of mAbs only neutralized hMPV. We named this antigenic site site III' (Fig. 4a, b). (2) The second group of mAbs mostly competed with two mAbs within its own group. A representative mAb in this group, M2A05, only neutralized the hMPV B subtype. To further map its epitope, we tested M2A05 binding to a panel of hMPV B F variants that converted subtype B-specific residues to their counterparts in hMPV A F sequence and found that a single D296K mutation of F antigen significantly reduced the binding of M2A05 (Supplementary Fig. 7B), suggesting that D296 is within the epitope of M2A05. This antigenic site, which sits below classical site IV on the hMPV F structure, was named as site IV' (Fig. 4a, b). (3) The third group of mAbs competed with M8C10, a mAb that likely targets a dynamic epitope on the internal trimerization interface (X Xiao, Z Wen, Q Chen, JM Shipman, J Kostas, JC Reid, A Tang, Z Chen, KA Vora, L Zhang, HP Su, MJ Eddins, manuscript submitted), suggesting that they all likely target internal epitopes. (4) The last group, named as M2E04-like mAbs by the most potent nAb in this group, did not compete with any of the other six groups mapped on the hMPV PreF surface. Most of M2E04-like mAbs only competed with mAb M1A08 within this same group, except M2E11L that competed with both M8C10 and M1A08. We were not able to determine the detailed epitopes of M2E04-like mAbs by epitope mutants or structural information, thus the epitope of M2E04-like mAbs remains to be explored. Interestingly, we did not find concrete evidence showing any mAb that targets site Ø. In the mAb group with high neutralizing potency (IC50 < 50 ng/mL),

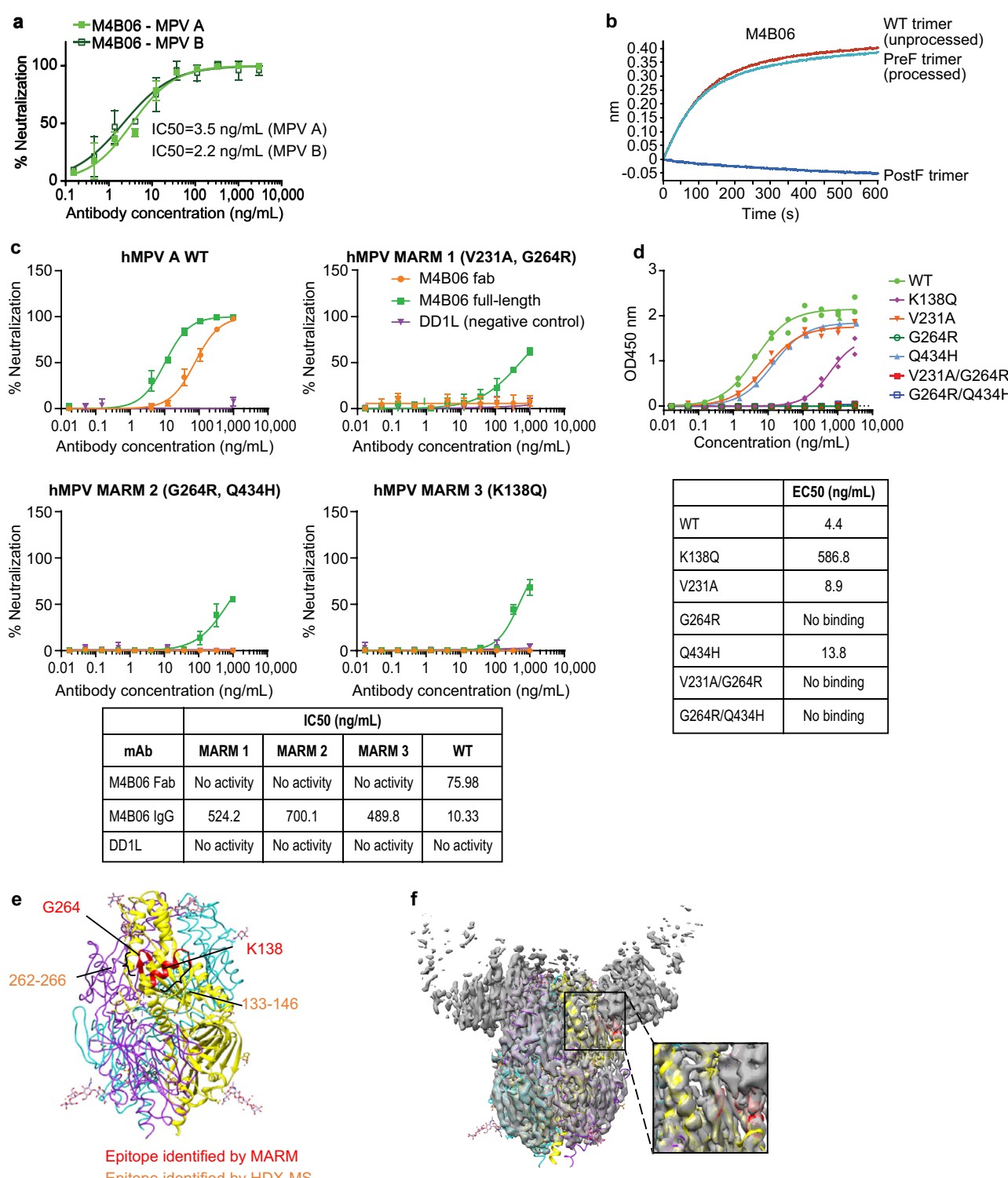

**Fig. 2 Characterization of an ultra-potent neutralizing mAb M4B06. a** Neutralization of M4B06 IgG to hMPV A and B strains. Data were presented as mean values ± SD of three replicates. **b** BLI binding of M4B06 IgG to unprocessed PreF trimer, processed PreF trimer, and PostF trimer. **c** Neutralization of M4B06 IgG and Fab, as well as an irrelevant control antibody (DD1L), to hMPV PreF and three types of MARM viruses. Data were presented as mean values ± SD of three replicates. **d** Binding of M4B06 IgG to unprocessed hMPV PreF WT and PreF carrying MARM mutants, determined by ELISA with Expi293 cell culture supernatants containing expressed antigens. Data were presented as mean values ± SD of two replicates. **e** M4B06 epitopes identified by MARM and HDX-MS were mapped on an hMPV PreF structure, with epitope residues colored in red in the ribbon diagram. **f** Cryo-EM map of three M4B06 Fabs in complex with a processed PreF trimer at 3.75 Å. The hMPV PreF model with the same coloring as (E) fit in the density. Source data are provided as a Source Data file.

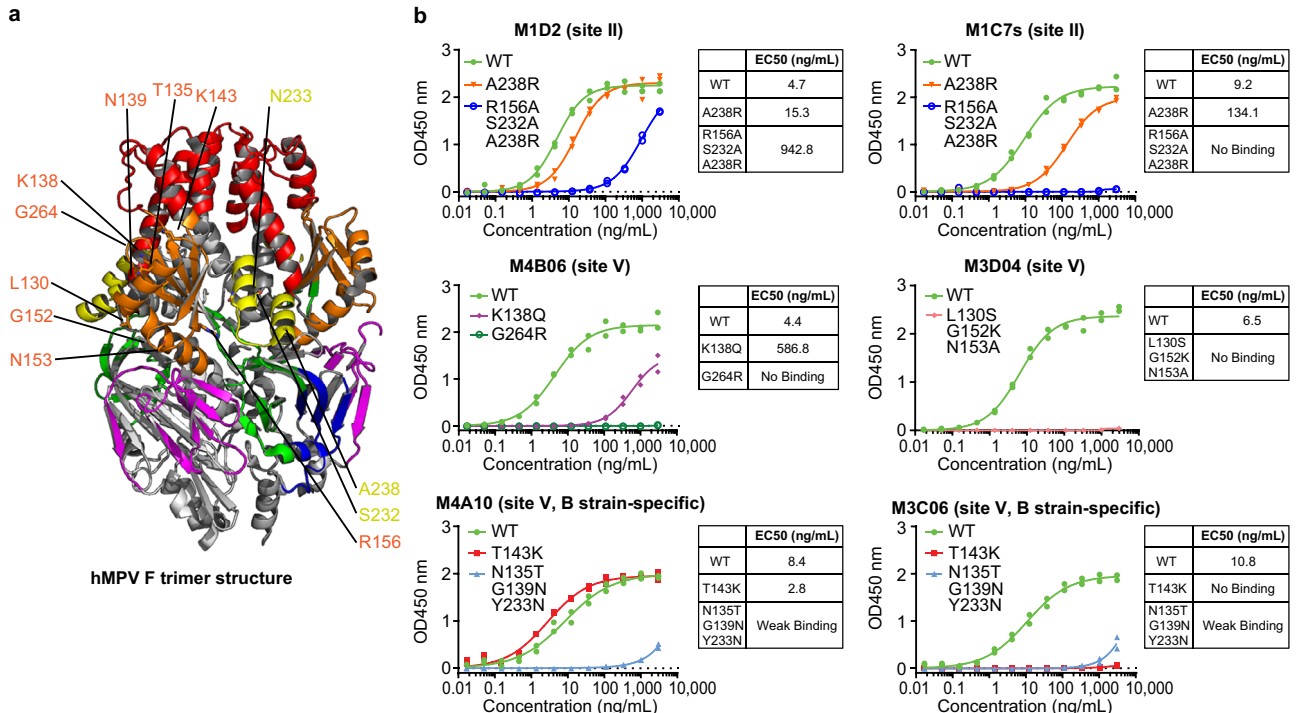

**Fig. 3 Epitope mapping of selected site II and V hMPV F-specific mAbs by binding to mutant F proteins. a** Contact residues for different site II (yellow) and V (orange) mAbs were mapped on an hMPV PreF trimer structure (PDB: 5WB0, subtype A)[28]. **b** Binding of selected site II and V mAbs to unprocessed hMPV PreF and mutants, determined by ELISA with Expi293 cell culture supernatants containing expressed antigens. Dots indicate two replicates. Source data are provided as a Source Data file.

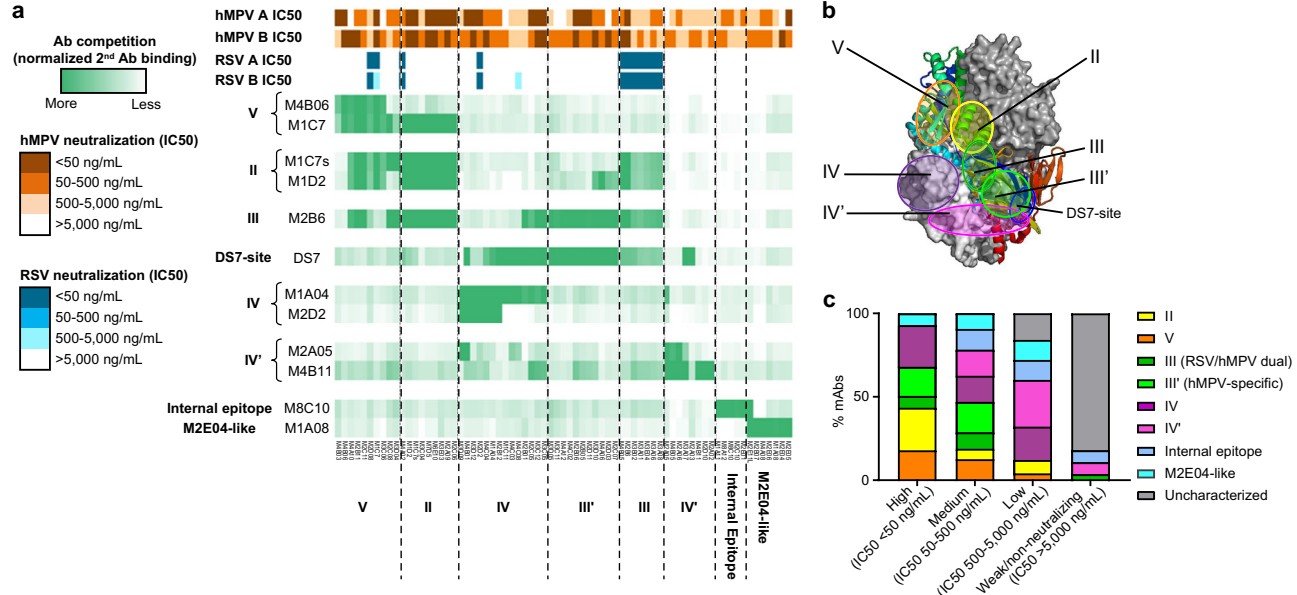

**Fig. 4 Epitope mapping of hMPV F-specific mAbs by BLI binning. a** Epitope binning of selected mAbs with the unprocessed hMPV PreF antigen. Seventy-three isolated mAbs which showed apparent BLI binding response (>0.2 nm) with the tested antigen were shown. The heatmap shows epitope binning from the sandwich-based BLI binding assay, with darker green color indicating more competition between the first and the second antibodies. Top side-bars showed the neutralization potency (IC50) of mAbs to hMPV A, hMPV B, RSV A, and RSV B virus strains. **b** Major antigenic sites II, III', IV, IV', V were mapped on the hMPV PreF trimer protein structure. DS7-site was also labeled as a reference. **c** Percentage of antibodies targeting each antigenic site, grouped by neutralization potency. For each mAb, the stronger neutralization potency of hMPV A and B were chosen for grouping. Source data are provided as a Source Data file.

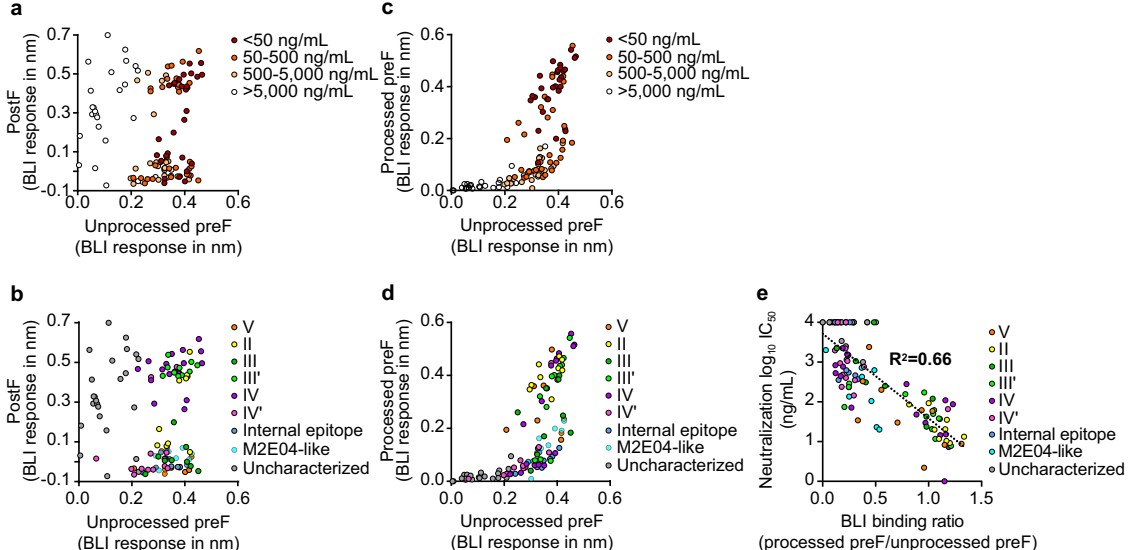

**Fig. 5 Correlations among neutralization potency, antigenic sites, and binding-specificity to different hMPV F conformations. a**, **b** BLI binding response of hMPV F-specific mAbs to unprocessed hMPV PreF trimer and PostF trimer. Each dot represents an isolated hMPV F-specific antibody, colored by neutralization potency (**a**) or mapped antigenic site (**b**). **c**, **d** BLI binding response of hMPV F-specific mAbs to unprocessed hMPV PreF and processed stabilized hMPV PreF (115BV) antigens. Each dot represents an isolated hMPV F-specific antibody, colored by neutralization potency (**c**) or mapped antigenic site (**d**). **e** Correlations between neutralization potency (after $Log_{10}$ transformation) and the ratio of processed PreF/unprocessed PreF BLI binding. The dashed line and R-square number indicate the linear regression. Each dot represents an isolated hMPV F-specific antibody, colored by the mapped antigenic site. For each antibody, the higher neutralization potency between hMPV A and B was chosen to plot. Source data are provided as a Source Data file.

less than 20% of mAbs targeted site V (Fig. 4c), and 67% of mAbs targeted sites II, IV, and III', which are located on the side of the hMPV F antigen (Fig. 4b). This is different from previous reports on RSV F which demonstrated immune dominance of sites Ø and V[36,48].

**Correlations among antigenic sites, neutralization potency, and binding-specificity to different hMPV F conformations.** To investigate the binding-specificity of isolated mAbs to different conformations of hMPV F antigens, we measured the binding-specificity of isolated mAbs to three forms of hMPV F antigens by BLI: (1) the uncleaved hMPV PreF trimer[34], which was used for B cell sorting and epitope binning (unprocessed PreF, Supplementary Fig. 2); (2) the prefusion-stabilized hMPV F trimer 115BV, which showed efficient cleavage between F1 and F2 after co-expression with furin[28] (processed PreF, Supplementary Fig. 2); and (3) an hMPV PostF, as confirmed by EM negative staining (Supplementary Fig. 8). Due to the high avidity between trimeric antigens and bivalent antibodies, the dissociation of many antibodies was too slow to determine an accurate Kd. Therefore, although the dissociation rates were not determined, we used the absolute binding response (nm) as an indicator of apparent binding-specificity as it reflects the association rates and epitope accessibility.

Comparing binding of isolated mAbs to unprocessed PreF with PostF showed that there were three groups of mAbs: PreF-specific, PostF-specific, and Pre/Post-dual mAbs (Fig. 5a). The groups of PreF-specific and Pre/Post-dual mAbs appear to have similar numbers of potent neutralizing mAbs (Fig. 5a). For example, in the 28 mAbs with neutralizing IC50 < 50 ng/mL and 33 mAbs with neutralizing IC50 between 50–500 ng/mL, 14 and 15 of mAbs were PreF-specific (defined as PostF binding response <0.3), respectively, suggesting that potent neutralizing mAbs could be elicited by both PreF-specific epitopes as well as epitopes presented on both PreF and PostF. Consistent with this data, we have also shown that PostF was able to deplete most of the

binding reactivity to PreF in these human donors' serum samples (Supplementary Fig. 9 and Supplementary Table 1). The PreF/PostF binding-specificity of mAbs was also well-correlated with their antigenic sites: site III' and site IV mAbs were mostly Pre/Post-dual binding antibodies, site II mAbs had a similar number of PreF-specific and Pre/Post-dual mAbs, and the rest of the other sites including site V, site III, site IV', site of the internal epitope (M8C10-like), and M2E04-like site, were PreF-specific (Fig. 5b).

We also compared the binding of isolated mAbs to unprocessed PreF with processed PreF. Interestingly, there are two apparently different groups of mAbs. One group of mAbs showed similar binding responses to both conformations of PreF; whereas the other group showed significantly reduced binding responses to processed PreF than to unprocessed PreF (Fig. 5c, d). This suggests that there are two types of epitopes—one type of epitope, enriched in site II, are presented on both processed and unprocessed PreF, (Fig. 5d and Supplementary Fig. 10); and the other group of epitopes, enriched in site IV', site of the internal epitope (M8C10-like), and M2E04-like site, are presented better in the unprocessed PreF, (Fig. 5d and Supplementary Fig. 10). Site II, III, III', and IV appear to include epitopes belonging to both groups (Fig. 5d and Supplementary Fig. 10). Further examination of relationships between binding and neutralization potencies demonstrated that the mAbs of the latter group with preferred binding to unprocessed PreF showed less neutralization potency than the mAbs of the former group (Fig. 5c and Supplementary Fig. 10). The neutralization potency of mAbs was negatively correlated with the ratio of processed PreF/unprocessed PreF binding (Fig. 5e), suggesting the potential differences in immunogenicity between unprocessed and processed hMPV F antigens.

**Characterization of the thermostability of unprocessed and processed hMPV F antigens.** To further characterize the differences between unprocessed and processed hMPV PreF antigens, we investigated their thermostability by differential scanning

**Table 2 Melting temperature (Tm) of unprocessed and processed hMPV F trimeric antigens determined by differential scanning fluorimetry (DSF).**

| Sample | Transition 1 (°C) | Transition2 (°C) |
|---|---|---|
| Unprocessed hMPV PreF trimer | 56.70 ± 0.33 | 83.86 ± 0.26 |
| Unprocessed stabilized hMPV PreF trimer | 55.50 ± 0.42 | n/a |
| Processed stabilized hMPV PreF trimer | 62.45 ± 0.09 | n/a |

fluorimetry (DSF). The unprocessed hMPV PreF construct, which does not include the A185P stabilizing mutation (Supplementary Fig. 2A), showed two unfolding transitions with the first transition at 56.7 °C and an additional transition at 83.9 °C (Table 2 and Supplementary Fig. 11A). In contrast, the melting temperature of furin processed stabilized hMPV PreF 115BV that contains the A185P mutation, as well as the RR furin-cleavage site at position 82 (Supplementary Fig. 2A), is significantly higher at 62.5 °C (Supplementary Fig. 11B), suggesting increased thermostability of the latter protein. To determine whether the improved thermostability is attributed to the introduction of A185P stabilizing mutation or furin-mediated proteolytic cleavage, we further generated unprocessed stabilized hMPV PreF 115BV protein in the absence of co-expressed furin (Supplementary Fig. 2A). The melting temperature of this A185P stabilized construct that had not been processed by furin is 55.5 °C (Supplementary Fig. 11C), similar to the unprocessed hMPV PreF protein without the A185P mutation, indicating that the furin-mediated proteolytic cleavage significantly improved the thermostability of hMPV PreF protein. Interestingly, the second high-temperature transition could only be observed for the hMPV F construct that does not include the A185P stabilizing mutation. We speculate that the addition of the stabilizing A185P mutation shifts this high-temperature transition to a temperature above 95 °C, which falls outside the observed temperature range of this experiment.

## Discussion

Despite the highly similar structural fold and some conserved epitopes between RSV and hMPV F proteins, whether the epitopes and potencies of neutralizing antibodies against hMPV F are similar to those of the RSV F mAbs remains unclear to date. In this study, we isolated and characterized over one hundred mAbs from multiple healthy adult donors to understand the antigenic sites of hMPV F-specific mAbs elicited from natural infection. Most isolated mAbs had significant somatic hypermutations compared with the germline sequences (Fig. 1c and Supplementary Fig. 3B, C), suggesting that the donors, which were pre-screened with hMPV F neutralizing antibody titers (Supplementary Fig. 12), were likely subjected to repeated hMPV infections in the past. Epitope binning and characterization of binding with F epitope mutants suggested that there are at least eight distinct antigenic sites of hMPV PreF, including multiple sites not previously reported (Fig. 4). We found that the numbers of isolated mAbs of each site vary by donor (Supplementary Fig. 12), possibly due to differences in their immune background. Nevertheless, this did not interfere with our analysis to characterize the common features of these mAbs regarding their targeted antigenic sites. A panel of mAbs, including M4B06, exhibited high neutralization potency in the in vitro neutralization assay with IC50 < 50 ng/mL, highlighting their potential applications as prophylactic or therapeutic interventions against hMPV infection. These mAbs were mapped to sites II, IV, V, III, and a newly-identified site III′ (Table 1). Within the pre-defined antigenic site II and V, multiple binding modes have also been identified (Fig. 3). A previously reported 'site IIIa' hMPV-specific nAb MPV364 does not compete with DS7 (DS7 site), 101 F (site IV), but competes with MPE8 and 25P13 (site III)[52], suggesting that MPV364 has a similar epitope binning profile to the "site II" mAbs reported in the current study. Further characterization and structural studies of MPV364 and the site II mAbs we reported here will help us understand the similarity and differences between these two groups of antibodies. In addition, we identified four antigenic sites that were not previously defined by the reported epitopes of RSV F (Fig. 4). Sites III′ and IV′, which are adjacent to previously defined sites III and IV, induced distinct groups of antibodies. Antibodies against two other antigenic sites, one located in the trimer interface (M8C10-like) and another yet to be characterized (M2E04-like), also comprised a significant portion of the total antibodies discovered. During the preparation of this manuscript, another hMPV mAb MPV458 was reported to target the 66–87 helix that is located near the trimer interface of the hMPV PreF apex region[53]. The epitope of MPV458 does not overlay with the epitope of M8C10, which is completely buried within the MPV F trimer (X Xiao, Z Wen, Q Chen, JM Shipman, J Kostas, JC Reid, A Tang, Z Chen, KA Vora, L Zhang, HP Su, MJ Eddins, manuscript submitted), suggesting multiple different epitopes are present within the trimerization interface. Whether MPV458 targets a similar epitope to the M2E04 group remains unclear and warrants further comparison.

Natural RSV infection elicits antibodies preferably targeting prefusion-specific apex sites such as sites Ø and V[48]. In contrast, for hMPV antibodies, analysis of human donors' serum samples in our study suggested that PostF was able to deplete most of the binding reactivity to PreF (Supplementary Fig. 9 and Supplementary Table 1). Moreover, we found comparable numbers of hMPV PreF-specific memory B cells vs. PreF/PostF-dual memory B cells and didn't observe the immune dominance of PreF-specific antigenic sites Ø and V (Fig. 4c), consistent with a previous report that the apex of hMPV F is heavily shielded by the glycans and less accessible for antibody recognition[28]. Interestingly, we have not isolated monoclonal antibodies that target the equivalent RSV PreF site Ø, which is on the apex of the PreF trimer. While it is possible that we were not able to obtain such antibodies due to technical limitations, we hypothesized that these antibodies might be rare due to the additional glycosylation in this area on hMPV PreF. The hMPV PreF and PostF antigens were reported to elicit comparable neutralizing antibody titers in mice[28], likely driven by PreF/PostF-dual binding nAbs as characterized in this study. The mAbs recognizing epitopes that are presented on both PreF and PostF might still be able to block the structural rearrangement from PreF to PostF, potentially through structural hindrance that prevents the PreF to PostF structural rearrangement outside of the epitope, therefore inhibiting the virus-cell fusion.

Since nascent hMPV F protein requires proteolytic activation by transmembrane proteases such as TMPRSS2 at the cell surface or in the virus particle[26], we hypothesized that both uncleaved and cleaved forms of PreF antigens may be accessible to humoral immunity. By examining the binding-specificity of isolated mAbs to both proteolytically unprocessed and processed PreF forms, we identified groups of mAbs that had an apparent binding preference for unprocessed PreF over-processed PreF, suggesting the potential differences in immunogenicity between the two PreF forms, despite their highly similar structures in monomeric forms. One explanation for this observation is that the proteolytic cleavage generates a hydrophobic end in the N-terminus of F1 segments, forming a hydrophobic core and stabilizing the trimerization of F protein (Supplementary Fig. 13). Indeed, DSF results confirmed that the proteolytic process improved the thermostability of the hMPV PreF trimer (Table 2 and

Supplementary Fig. 11). The unprocessed PreF trimer might present a more dynamic conformation, therefore exposing some epitopes which are less accessible on the processed and more stabilized PreF trimer. For example, a group of unprocessed PreF-preferred mAbs competed with M8C10, a mAb recognizing an internal epitope within the trimerization interface (X Xiao, Z Wen, Q Chen, JM Shipman, J Kostas, JC Reid, A Tang, Z Chen, KA Vora, L Zhang, HP Su, MJ Eddins, manuscript submitted). Similar mAbs have been reported for other viruses such as influenza[57]. One caveat of our study is that only unprocessed hMPV PreF was used as the bait for antibody isolation due to the availability of reagent when we started our study, thus we would not be able to identify mAbs targeting epitopes that are only present on the processed hMPV PreF. To address this concern, we examined the cross-reactivity of different hMPV F conformations by a serum absorption assay. We found that unprocessed PreF was able to deplete most of the cross-reactivity to processed PreF (Supplementary Fig. 9 and Supplementary Table 1), suggesting that mAbs binding exclusively to processed PreF do not represent a significant portion of serum hMPV F-binding antibodies. However, it is still possible that analysing memory B cells with the processed PreF antigen could lead to the discovery of antibodies specific to processed PreF, and it will be an interesting subject for future studies. On the other hand, processed PreF was also able to deplete cross-reactivity to unprocessed PreF in the serum (Supplementary Fig. 9 and Supplementary Table 1), likely because the mAbs with unprocessed PreF-preferred binding still retain weak binding to processed PreF antigen when the antigen is excessive (Fig. 5c, d). It is interesting that there was a negative correlation between the in vitro neutralization potency and the binding preference to the unprocessed F than the processed F, suggesting that processed PreF trimer may serve as a better immunogen than unprocessed PreF for eliciting higher neutralizing antibody response. However, it is worth noting that there may be other potential antiviral mechanisms independent of virus neutralization. For example, the previously reported FluA-20 antibody did not show in vitro neutralization activity but was able to inhibit in vitro cell-cell fusion and provide in vivo protection in mice[57]. Therefore, further investigation is needed to better understand the protective roles of such unprocessed PreF-preferred mAbs and whether their epitopes are important to be included in hMPV vaccine design.

Taken together, the characterization of a large panel of hMPV F mAbs isolated from natural infection in our present study revealed the humoral immune recognition that differs from RSV F and provided insights into future structural-based design and development of hMPV antibodies and vaccines for prophylactic and therapeutic interventions.

## Methods

**Ethics statement**. Blood samples from healthy adults were collected with informed written consent obtained in accordance with the Helsinki Declaration of 1975 (approved by the Institutional Review Board of Merck & Co., Inc., Kenilworth, NJ, USA).

**Production of recombinant proteins**. The plasmid construction and production of RSV PreF (DS-Cav1) was performed as previously described[31,58]. The unprocessed hMPV PreF trimeric antigen was derived from a previously published F sequence of strain B2 with a C-terminal GCN4 trimerization domain, which adopted a PreF-like structure[34]. The monomeric hMPV F antigen was derived from the same B2 strain but without the GCN4 domain. The stabilized hMPV PreF construct (115BV) adopted the prefusion stabilizing mutations (including a proline mutation A185P and insertion of a furin-cleavage site) from a previous report[28] on the backbone of wild-type strain B2 hMPV F trimeric antigen with C-terminal GCN4 domain (Supplementary Fig. 2). The hMPV postF sequence was derived from strain B2, where a furin-cleavage site (RKRR) was introduced to the original F1/F2 cleavage site, the fusion peptide sequence FVLGAIAL at the N-terminal of F1 was deleted, and a C-terminal foldon trimerization domain was added. The hMPV F mutants were made on the backbone of the unprocessed hMPV F trimeric

antigen. All constructs have a C-terminal thrombin cleavage site, followed by a 6xHis tag and a strep-tag. Sequences were codon-optimized for mammalian expression (Life Technologies and Genewiz), cloned into an expression vector, and transiently transfected into Expi293 suspension cells (Life Technologies). The processed stabilized hMPV PreF construct (115BV) was co-transfected with furin at a 1:1 ratio, and other hMPV F constructs were transfected without furin. On days 3 to 7 post-transfection, supernatants were harvested for western blot to confirm expression, for direct ELISA binding assay, and for large-scale purification. The purification of all antigens was performed as previously described[58]. Briefly, harvested supernatants with His-tagged proteins were captured by Ni-Sepharose chromatography (GE Healthcare) and eluted by high imidazole concentration. After overnight dialysis in the presence of thrombin, the His-tag was cleaved, and the concentration of imidazole was reduced. Uncleaved His-tag products, as well as initial Ni-Sepharose non-specific binding impurities, were removed by negative Ni-Sepharose chromatography (product in flow-through). The protein antigens were further purified by size-exclusion chromatography (Superdex 200, GE Healthcare) and stored in a buffer of 50 mM HEPES pH 7.5 with 300 mM NaCl.

**Human subjects and PBMC preparation**. Blood samples from healthy adults were purchased from Biological Specialty Company (Colmar PA) after written informed consent. Plasma or serum samples from these donors were screened for activity in an hMPV microneutralization assay. Peripheral blood mononuclear cells (PBMCs) were purified from blood collected in EDTA tubes by density gradient centrifugation in histopaque over Accuspin™ tubes (Sigma Aldrich) according to the manufacturer's instructions. PBMCs were then frozen in 90% heat-inactivated FBS supplemented with 10% dimethyl sulfoxide and stored in liquid nitrogen until thawed for use in assays.

**Antigen-specific single memory B cell sorting and culture**. Human memory B cells were single-sorted with specific antigens as previously described with minor modifications[56,59]. PBMC from selected donors were isolated from whole blood on the same day as blood collection, using an Accuspin tube (AccuspinTM System-Histopaque®-1077 (Sigma Cat#A6929). Cells were then frozen in 90% Fetal calf serum with 10% DMSO, and stored in liquid nitrogen until thawed for experiments. The purified recombinant unprocessed hMPV PreF trimer and monomer proteins were biotinylated (Thermo) and used as sorting antigens (Supplementary Data 1, column D). For a portion of the experiments, B cells were sorted using a mixture of biotinylated unprocessed hMPV PreF and RSV PreF (DS-Cav1) tagged with Alexa-647 (Supplementary Data 1, column D). Cryopreserved PBMCs were thawed on the day of sorting and the B cell population was enriched using the EasySep™ Human B-cell Enrichment Kit (Stemcell Technologies). Next, B cells were stained with biotinylated or Alexa-647 tagged F antigens and then followed by staining with a panel of monoclonal antibodies including anti-CD3 mAb-PE-Cy™7 (BD Biosciences Cat#563423, 1:60 dilution), anti-CD19-FITC (BD Biosciences Cat#555415, 1:15 dilution), anti-human IgG-APC (BD Biosciences Cat#550931, 1:15 dilution), and PE-streptavidin. CD3−/CD19+/IgG+/F+ cells were sorted with a SONY 800 S Cell Sorter in single-cell mode into a 96-well plate (Supplementary Fig. 1B, FACS plots were analyzed by SONY Cell Sorter Software v2.1.5). The sorted memory B cells were then cultured for 14 days at 37 °C, 5% $CO_2$ for conversion into IgG secreting cells as previously described[59]. At the end of the cultures, plates were centrifuged at 935×g for 10 min. The culture supernatants were then transferred to new 96-well plates and screened for ELISA positive binding to the hMPV PreF trimer. The cell pellets were lysed in 50 μL of RLT buffer (Qiagen) supplemented with 1% 2-mercaptoethanol (Sigma) and stored at −80 °C for immunoglobulin gene sequencing.

**Recovery of antibody sequences and antibody production**. The antibody sequences from the wells with positive ELISA binding to hMPV F were further recovered by regular cloning method as previously described[59] or by barcode-based next-generation sequencing (NGS, Supplementary Fig. 1A). Specifically, for NGS cloning, after RNA extraction and the first step of RT-PCR to amplify variable regions, 5′ and 3′ overlapping sequences were attached by the second step of multiplex nested PCR, followed by a third-step of PCR to add 5′ and 3′ barcode and adapter sequences. The final amplicons, which contain unique 5′ and 3′ barcode combinations for each well, were pooled together and sequenced by Illumina MiSeq. Primer sequences were included in Source Data.

NGS sequences were checked for quality by FastQC v0.11.2 (bioinformatics.babraham.ac.uk/projects/fastqc) and paired-end reads were assembled using PANDAseq v2.10[60] with a minimum overlap region of 10 bp. The assembled BCR reads were aligned to V(D)J germline sequences (IMGT v3.1.19)[61] using IgBLAST v1.9.0[62] and to IGHC sequences using tblastn (v2.2.29) as a part of the BLAST + suite[63]. The output of IgBLAST was processed using commands from Change-O v0.3.12[64]. Only functional sequences with an E-value of no more than 0.001 for the alignment of the V and J genes were retained for downstream analysis. The most abundant sequence obtained from each well was representative of the antibody sequence recovered from the well.

The naturally paired heavy and light chain variable region sequences obtained from single-sorted human memory B cell cultures were sub-cloned into pTT5 vector for expression in CHO-3E7 cells (performed at GenScript). Briefly,

CHO-3E7 cells were grown in serum-free FreeStyle CHO Expression Medium (Life Technologies). The recombinant plasmids encoding heavy and light chains of each antibody were transiently co-transfected into suspension CHO-3E7 cell cultures. The culture supernatants collected were used for purification with a Protein A CIP column (GenScript). The purified antibodies were QC tested by SDS-PAGE and western blot.

**hMPV plaque reduction neutralization assay**. A plaque reduction neutralization assay was developed for hMPV similarly as described previously for RSV[65]. Briefly, antibodies in serial dilutions with OptiMEM medium (Gibco) were first added into Poly-D coated 96-well flat-bottom plates (Corning Costar) at 50 μL per well. For initial screening, the hMPV A1 and B2 viruses (ZeptoMetrix Corp) at 2000 pfu/mL were mixed with antibodies at 50 μL per well and incubated for 1 h at 37 °C with 5% $CO_2$. The LLC-MK2 cells (ATCC Cat#CCL-7.1) at $0.8 \times 10^6$ to $1.2 \times 10^6$ cells/mL in OptiMEM medium were then added to the antibody/virus mixtures at 25 μL per well. After 1 h incubation at 37 °C with 5% $CO_2$, the plates were centrifuged at $336 \times g$ for 10 min. About 125 μL of OptiMEM medium supplemented with 1% methylcellulose was overlaid in each well. Plates were incubated at 37 °C with 5% $CO_2$ for 4 days. Cells were then fixed with 10% formalin (Fisher Scientific) at 100 μL per well for 30 min at room temperature. The plates were dried for 20 min before washing with PBST. Fixed cells were blocked with blocking buffer (Odyssey) for 30 min, followed by 2 h incubation with a mouse anti-hMPV mAb clone 132 (EMD Millipore MAB80124, 1:1000) diluted in blocking buffer at 50 μL per well. Plates were washed again by PBST and then incubated with an anti-mouse IgG Alexa 488 conjugated secondary antibody (Invitrogen #A11017, 1:500) in assay diluent at 50 μL per well for 1 h. After washing off the excessive secondary antibodies by PBST, the plates were read and counted by EnSight (PerkinElmer). The IC50 was calculated from 4 parameters nonlinear fitting algorithm using GraphPad Prism 8 software. For the antibodies that showed an IC50 < 10 μg/mL in the initial screening, neutralization assays with the same procedure were repeated in a wider range of antibody concentrations with hMPV A2 (RL_bx from Baylor College of Medicine) and B2 (Peru6-2003) viruses.

**RSV plaque reduction neutralization assay**. A plaque reduction neutralization assay was developed for RSV as described previously[65]. Briefly, antibodies in serial dilutions with EMEM medium supplemented with 2% FBS and 2 mM glutamine were first added into Poly-D coated 96-well flat-bottom plates (Corning Costar) at 50 μL per well. The RSV A Long and B Washington strains at 2000 pfu/mL were mixed with antibodies at 50 μL per well and incubated for 1 h at 37 °C with 5% $CO_2$. The HEp-2 cells (ATCC, cat#CCL-23) at $0.8 \times 10^6$ to $1.2 \times 10^6$ cells/mL in EMEM medium supplemented with 2% FBS and 2 mM glutamine were then added to the antibody/virus mixtures at 25 μL per well. After 1 h incubation at 37 °C with 5% $CO_2$, the plates were centrifuged at $336 \times g$ for 10 min. Each well was overlaid with 125 μL of EMEM medium supplemented with 2% FBS, 2 mM glutamine, and 1% methylcellulose. Plates were incubated at 37 °C with 5% $CO_2$ for 3 days. Cells were then fixed with ice-cold 80% acetone (Sigma) in PBS at 100 μL per well for 10–20 min at room temperature. The plates were dried for 20 min before washing with PBST. Fixed cells were stained with a mixture of an in-house generated mouse anti-F monoclonal antibody (clone #143-F3-B138, 1.25 μg/mL) and an in-house generated mouse anti-N (nucleoprotein) monoclonal antibody (clone #34C9, 1.25 μg/mL) from hybridoma. Plates were washed again by PBST and then incubated with an anti-mouse IgG Alexa 488 conjugated secondary antibody (Invitrogen #A11017, 1:500) in assay diluent at 50 μL per well for 1 h. After washing off the excessive secondary antibodies by PBST, the plates were read and counted by EnSight (PerkinElmer). The IC50 were calculated from four parameters nonlinear fitting algorithm using GraphPad Prism 8 software.

**Cryo-EM study of M4B06 and hMPV F complex**. Cryo-EM sample preparation and density map determination were performed by Wuxi Biortus Biosciences (Wuxi, China). For sample preparation, the M4B06-hMPV PreF complex was prepared by incubating recombinantly purified M4B06 Fab with processed hMPV PreF (115BV) trimer (PreF trimer: Fab = 1:3) for 1 h at 4 °C, followed by purification using Superdex 200 Increase 10/300 GL column in the buffer of 25 mM HEPES pH 7.5, 150 mM NaCl. The fraction sample was flashed frozen and pre-checked with negative staining for a subsequent cryo-EM study. Three microliters of the purified sample at 0.2 mg/mL were applied onto a glow-discharged holey carbon film grid with Graphene Oxide (Quantifoil, 300 mesh, R2.0/1.0). Samples were blotted for 6 s and vitrified by plunging into liquid ethane cooled by liquid nitrogen using a Vitrobot Mark IV (Thermo Fischer Scientific) at 4 °C and 100% humidity.

For Cryo-EM data acquisition, a total of ~1800 movie micrographs (36 frames for each movie) were acquired on the FEI Titan Krios 300KV electron microscope using a Gatan K3 Summit direct electron detector, operated in super-resolution mode applying a total dose of 75 $e^-/Å^2$ during a 3 s exposure. Each micrograph was acquired (with SerialEM software) at 89,000x nominal magnification (0.5225 Å/pixel at the specimen level) with a nominal defocus range of −1.5 to −3.0 μm.

For image processing and 3D reconstruction, all movie frames were aligned and binned by a factor of 2 using RELION 3.0[66] and their CTF parameters were estimated with Gctf[67]. Approximately 1000 particles were manually boxed and extracted from ~30 far-defocus micrographs, and their 2D average classes were used as templates to automatically box particles from all selected micrographs with autopick in RELION 3.0[68]. A total of ~1,000,000 picked-particle images were extracted (binned by a factor of 8 to give a pixel size of 4.18 Å) and subjected to reference-free 2D alignment with RELION 3.0. After multiple iterations of 2D classifications, particles (~218,000) belonging to good classes were 3D classified against the map created from manually picked particles as the initial model. After two iterations of 3D classifications, particles (~131,000) in good classes were re-centered, re-extracted (binned by a factor of 5 to give a pixel size of 2.6 Å), and refined to yield a map at Nyquist frequency of 5.2 Å resolution. Using this map as a model, particles were further 3D classified into six classes (Supplementary Fig. 6A) without any further origin and orientation determination. Particles (87,876) in those three classes showing better identifiable structural features were re-extracted (binned by a factor of 2 to give a pixel size of 1.045 Å) and further refined to a final map with a resolution of 3.75 Å (FSC = 0.143) without imposing any symmetry.

For model building, the model of an hMPV PreF trimer of B strain was built using the crystal structure of A strain as the template[28]. The M4B06 Fab was not built in the density map. The model was docked into the Cryo-EM map using Chimera[69]. Structural figures were prepared in Chimera.

**Transmission electron microscopy and image analysis**. Transmission electron microscopy and 2D class averaging were performed by NanoImaging Services, Inc. (San Diego, CA). Samples were prepared on continuous carbon films supported by nitrocellulose-coated 400 mesh copper grids (Ted Pella). A 3 μl drop of purified hMPV PostF protein at a concentration of 2–8 μg/mL was applied to a freshly plasma-cleaned grid for 1 min and blotted to a thin film using filter paper, followed by immediate staining with 1% (w/v) uranyl formate. Transmission electron microscopy was performed using an FEI Tecnai T12 electron microscope operating at 120 kV equipped with an FEI Eagle 4k x 4k CCD camera. Negative stain grids were transferred into the electron microscope using a room temperature stage. Images of each grid were acquired at multiple scales to assess the overall distribution of the specimen. After identifying potentially suitable target areas for imaging at lower magnifications, pairs of high magnification images were acquired at nominal magnifications of 110,000x (0.10 nm/pixel) and 67,000x (0.16 nm/pixel) using the automated image acquisition software package Leginon[70]. Images were acquired at a nominal underfocus of −2 to −1 μm and electron doses of approximately 30–35e/Å2.

Image processing was performed using the Appion software package[71]. Contrast transfer functions of the images were estimated using CTFFIND4[72]. Individual particles in the 67,000x or 110,000x high magnification images were selected using automated picking protocols[73], followed by several rounds of reference-free alignment and classification using the X-windows-based microscopy image processing package (Xmipp[74]), which contains an algorithm that aligns the selected particles and sorts them into self-similar groups of classes.

**Generation of monoclonal antibody-resistant mutants (MARMs) for M4B06**. Vero cells were seeded in six-well plates in OptiPRO SFM (Gibco) in supplement with 1% Pen Strep (Gibco) and 2% L-Glutamine (Gibco) at 37 °C with 5% $CO_2$. On the next day, the cells were infected by a mixture of hMPV A2 virus (RL_bx from Baylor College of Medicine) at 10 pfu/cell and M4B06 Fab at 2, 8, and 40 μg/mL in three rounds respectively. In every round, cells were cultured at 37 °C with 5% $CO_2$ and harvested when syncytia appeared (day 6), lysed by three times of freeze/thaw in liquid nitrogen, and centrifuged at 500 g for 10 min. One milliliter of supernatant containing virus was used to infect fresh monolayer Vero cells (ATCC Cat# CCL-81). The remaining supernatant was mixed with 10x SPGA buffer (Biological Industries) and frozen for long-term storage. After three rounds of culture and infection, the virus supernatant from the last round was prepared in tenfold serial dilutions and 0.5 mL of supernatant was used to incubate with fresh monolayer Vero cells at 37 °C with 5% $CO_2$. After 1 h incubation, 3.5 mL overlay (OptiMEM medium supplemented with 1% methylcellulose) was added, and cells were cultured at 37 °C with 5% $CO_2$ for a week. The plaques were identified based on the CPE. Twenty-two single plaques were then collected separately and inoculated in fresh monolayer Vero cells prepared in a 24-well plate for 5 days at 37 °C with 5% $CO_2$ before harvesting. The final harvested MARM viruses were lysed in RLT buffer (Qiagen) with 1% 2-mercaptoethanol and viral RNA were extracted using RNeasy kit (Qiagen). The F genes from individual plaques were amplified by RT-PCR using viral RNAs, followed by PCR with F-specific primers, gel extraction, and sequencing.

**Biolayer Interferometry (BLI)-based binding assay and epitope binning**. BLI-based assays were performed using an Octet Red 96e instrument (ForteBio, Inc.). All experiments were performed at 30 °C with shaking at 1000 rpm. For binding assay, recombinant unprocessed hMPV PreF trimer, processed stabilized hMPV PreF trimer, and hMPV PostF trimer were used as antigens. The antibodies and antigen proteins were diluted in the kinetics buffer (ForteBio) at concentrations of 0.5 and 2 μg/mL, respectively. The antibodies and a reference well with empty buffer were first individually immobilized on anti-human Fc capture (AHC) biosensors (ForteBio) for 180 s, followed by the association step with antigen for 600 s.

Most of the antibodies' association curves were within linear range without showing saturation. For data processing, the absolute binding response (nm) of each antibody sample was subtracted by the binding response (nm) of reference well by Octet software DataAquisition version 8. The plots were generated by GraphPad Prism 8 software.

For Epitope binning, 73 isolated mAbs which showed apparent BLI binding response (>0.2 nm) were included. Recombinant unprocessed hMPV PreF trimer was used as the antigen. All the protein samples were diluted in the kinetics buffer (ForteBio). The antibodies to be characterized or the reference mAbs to be used as internal controls (5 µg/mL) were first individually immobilized on anti-human Fc capture (AHC) biosensors (ForteBio) for 600 s, followed by blocking with an irrelevant mAb (5 µg/mL) for 600 s. The biosensors were then immersed in wells containing the F antigen (2 µg/mL). Finally, the antigen-loaded biosensors were immersed in wells containing the second reference mAbs (5 µg/mL). The primary binding responses of mAbs binding and antigen loading were obtained from Octet software DataAquisition version 8. The normalized secondary mAbs binding was calculated by the binding responses of second reference mAbs binding divided by the binding responses of antigen loading. The normalized numbers were then plotted as a clustered heatmap by heatmap.2 package in R Studio 1.2.5033. MAbs with similar competition profiles as the reference mAbs were clustered into the same categories. The neutralization IC50 values of hMPV A, B and RSV A, B were plotted as top bars in the heatmap (raw data in Source Data).

### Epitope mutant design and ELISA assay

The mutants were rationally designed with several different strategies: (1) Based on reported structures of RSV antibodies, we selected residues critical for RSV antibody binding, and mutated the equivalent residues and surrounding residues on hMPV F, assuming the binding mode of hMPV antibodies are similar to the RSV antibodies. We used this approach to design the site II mutations A238R and R156A/S232A/A238R, and site IV mutation D414K. (2) Based on previously reported RSV epitope mutations, we mutated counterpart residues on hMPV F. We used this approach to identify site V/AM14-like epitope mutations L130S/G152K/N153A. (3) Some antibodies only neutralize hMPV B but not hMPV A, so we mutated the hMPV F B-strain specific residues to A-strain specific residues to map these antibodies. We use this approach to identify site V mutations T143K, N135T/G139N/Y233N, and site IV' mutations D296K. (4) M4B06 epitope mutations were designed based on the MARM sequences.

For ELISA assays with unpurified antigens in Expi293 supernatants, 96-well Ni-NTA coated plates (Thermo Scientific) were coated with cell culture supernatants for 2 h at room temperature. Plates were then blocked by the addition of 2% (v/v) bovine serum albumin (BSA) in PBS. After the blocking step, plates were incubated with serial dilutions of antibodies at room temperature for 90 min. Plates were washed by PBST and then incubated with HRP-conjugated goat anti-human IgG (Southern Biotech #2040-05, 1:2,000) for 45 min. Plates were washed again and developed with TMB solution (Virolabs). Absorbance at 450 nm was read on a plate reader (Victor III; PerkinElmer). EC50 values were calculated with Hill slope curve fitting using GraphPad Prism 8 software.

### Epitope mapping of M4B06 by HDX-MS

To optimize the sequence coverage, several preliminary experiments using various quench conditions, digestion, desalting, liquid chromatography, and mass spectrometry parameters were evaluated. The following represents the conditions that were determined to be optimal for the hydrogen-deuterium exchange experiments. Stock solutions of hMPV prefusion-stabilized PreF trimer (115BV-GCN4) with and without M4B06 monovalent Fab were prepared in 1x PBS. The Ag:Ab complex was prepared by incubating hMPV F with M4B06 at ambient temperature for 1 h before cooling to 1 °C. The hydrogen-deuterium exchange reactions were initiated by adding 40 µL PBS prepared in D2O to 10 µL of the hMPV F and hMPV F/M4B06 solutions. The labeling reactions were performed at 1 °C for five-time points: 15, 50, 150, 500, and 5000 s. After these time points the samples were quenched by mixing 40 µL of the labeled solution with 20 µL of quench solution (2 M Urea, 1 M TCEP, adjusted to pH 3.0 with NaOH) at 1 °C. About 50 µL of the quenched sample was immediately injected onto an immobilized pepsin column (Waters Enzymate BEH Pepsin column, 2.1 × 30 mm) held at 10 °C for online proteolysis using 0.05% (v/v) TFA in water solvent at a flow rate of 200 µL/min. The flow, with the digest, continued at the same rate onto a trap column (Waters UPLC BEH 300 C18, 1.7 µm, 2.1 × 5 cm) held at 0 °C for desalting. The combined digestion and desalting steps lasted for 90 s after which the trap column flow was reversed and eluted onto an analytical column (Waters Acquity UPLC CSH C18, 1.7 µm, 1.0 × 100 mm) held at 0 °C using a gradient of solvent A: 0.05% (v/v) TFA in water and solvent B: 0.0025% (v/v) TFA in 95% acetonitrile/5% water (v/v). A linear gradient of 13 to 40% B over 9.5 min at a flow rate of 40 µL/min was employed to elute the peptides into a Thermo Scientific LTQ-XL Orbitrap mass spectrometer (Thermo Fisher Scientific) operated in positive mode at a resolution of 30,000. The data were acquired over the m/z range of 350–2000 in profile mode using the following MS parameters: HESI source voltage 4 kV, capillary voltage 39 V, tube lens 130 V, capillary temperature 250 °C, and sheath gas flow 10. The labeling, quench, injection, digestion, desalting, and elution steps were performed using an HTC PAL robot (LEAP Technologies) controlled by HDxDirector software. The isocratic and gradient solvent flows were obtained using Waters nano-Acquity UPLC pumps (Waters). Prior to the exchange experiment, the identity of

each peptide was confirmed by analyzing a non-deuterated hMPV sample in data-dependent MSMS mode and processed with Proteome Discoverer 1.4 software (Thermo Fisher Scientific). The HDExaminer software version 1.4 (Sierra Analytics) was used to determine the centroid mass of the isotopic envelope of each peptide in the labeling experiment and quantifying deuterium incorporation.

### Differential scanning fluorimetry for unprocessed and processed hMPV F proteins

Analyses were performed with unprocessed hMPV PreF trimer protein and processed stabilized hMPV PreF trimer protein expressed with and without furin that had been stored at −70 °C. Two-fold serial dilutions of hMPV F protein in 50 mM HEPES, 300 mM NaCl at pH 7.5 were prepared from starting concentrations of 1.5, 2.5, and 2.7 mg/mL, respectively, to obtain samples in a concentration range between 3.125 µM–50 µM. Differential scanning fluorimetry (DSF) measurements were performed using the Prometheus NT.48 instrument (NanoTemper Technologies GmbH, Munich, Germany). In brief, experiments were performed in duplicate using high sensitivity capillaries (PR-C006). Each capillary was filled with 10 µL of protein solution and the fluorescence signal at 350 and 330 nm was recorded at temperatures between 20 to 95 °C at a temperature increase rate of 1 °C/min. Instrument software was used to calculate the ratio between the 350 and 330 nm signal as well as the first derivative. Maxima of the first derivative were used to obtain transition midpoints of protein unfolding events.

### Serum absorption assay

Serum samples of four donors where the antibodies were isolated (#84974, #86559, #21083, and #85359) were tested in the absorption assay. Each sample (50 µL) is diluted in PBS to a total volume of 970 µL and mixed with 20 µL of 0.5 mg/mL antigen or PBS buffer control for 2 h incubation at room temperature. The samples were then mixed with 10 µL of reconstituted Strep-tag II mAb (GenScript) at 0.5 mg/mL and placed on a 360° rotator at 4 °C for 2 h. Pre-washed 80 µL of sheep anti-mouse immunoglobulin G Dynabeads (2 mg/mL, Life Technologies) were then added to the mixture for 1 h incubation at 4 °C on a rotator. Dynabeads carrying antigen and antigen-bound antibodies were separated by DynaMag-2 magnet (Life Technologies), leaving absorbed supernatant for further ELISA binding assay with F antigens. ELISA titers were calculated by dilution folds and averaged ELISA titers from four donors were plotted.

### Computational sequence and structural analysis

The V gene germline usage, clonal analysis, and somatic hypermutation (SHM) were analyzed by IgBlast[62] based on the Kabat delineation system[75]. The clonotype was defined to include any V(D)J rearrangements that have the same germline V(D)J gene segments, the same productive/non-productive status, and the same CDR3 nucleotide as well as amino sequence. Structural visualization was generated by PyMOL 1.7.05 (Schrödinger). The plots were generated by GraphPad Prism 8 software.

**Reporting summary**. Further information on research design is available in the Nature Research Reporting Summary linked to this article.

## Data availability

Source data used to generate figures are provided with this paper. The Cryo-EM map of M4B06-hMPV F has been deposited in EMDB (ID: EMD-25810). CDR3 sequences are provided in Supplementary Data 1. Complete antibody sequences reported in this manuscript have been filed in a patent application (US63/148920; US63/276172) and will be disclosed when it publishes. In the interim, antibody sequences are available under a material transfer agreement by contacting Kalpit A. Vora (kalpit.vora@merck.com), Zhifeng Chen (zhifeng.chen@merck.com), or Lan Zhang (lan_zhang2@merck.com).

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

## Acknowledgements

The authors would like to thank Drs. James C. Cook, Bonnie J. Howell, and MRL Postdoctoral Research Program for their support of this work, Drs. Yacob G. Llorente, Chuan Hong, and Zhe Wu for their suggestions on the Cryo-EM study, Dr. Pedro A. Piedra for providing hMPV viruses, and Drs. Amy S. Espeseth, Antonios O. Aliprantis, and Daria J. Hazuda for helpful discussions and review of this manuscript.

## Author contributions

K.A.V., Z.C., and Lan Z. conceived the project; X.X., A.T., K.S.C., and N.L.S. isolated the mAbs; X.X., Lu Z., and A.F. analyzed the antibody sequences and NGS data; X.X., J.D.G., S.C., and M.J.E. produced the proteins; X.X., P.P., E.D., M.M., Z.W., and R.M. characterized the mAbs and antigens; D.T. and X.Y. conducted the Cryo-EM study; A.J.B., H-P.S., A.F., K.A.V., Z.C., and Lan Z. supervised the project; X.X. drafted the manuscript. All authors reviewed the draft and approved the manuscript for publication.

## Competing interests

X.X., A.F., Lu Z., P.P., E.D., M.M., A.T., K.S.C., Z.W., R.M., J.D.G., S.C., M.J.E., N.L.S., A.J.B., H-P.S., K.A.V., Z.C., and Lan Z. are current or former employees or contractors of Merck Sharp & Dohme Corp, a subsidiary of Merck & Co., Inc. Kenilworth, NJ, USA and may hold stock in Merck & Co., Inc. Kenilworth, NJ, USA. In addition, a subset of the authors (X.X., A.T., K.S.C., Z.W., J.D.G., H-P.S. K.A.V., Z.C., and Lan Z.) are listed as inventors on a patent surrounding this work held by Merck Sharp & Dohme Corp. The remaining authors declare no competing interests.
