## [Peer Review File · Nature Communications]

Profiling of hMPV F-specific antibodies isolated from human memory B cellsReviewers' Comments:

Reviewer #1:

Remarks to the Author:

This is a solid paper that describes an antibody discovery campaign for human metapneumovirus, targeting the fusion protein. This is a logical target, given that previous literature on our RSV and MPV have shown that the F protein is the principal neutralizing target for this virus. A lot of work has been done in this field on RSV, but much less work has been done on MPV. MPV and RSV cross reactive antibodies also have been described. Therefore, this is a mature field, but this paper adds a lot more MPV F antibodies into the literature and thus, could be a substantial contribution. There are a lot of words like novel and 1st in the paper that are a bit of a stretch, and all that language could be removed from this paper, without denigrating the presentation of the data.

The approach is to use single B cell technologies to isolate naturally occurring human monoclonal antibodies from human donors who have been previously infected naturally. The investigators use common techniques of proteomics and escape mutants for epitope mapping to identify epitopes, or at least the major antigenic sites, identified by these antibodies. There is also a lot of work done on the recognition of pre- versus post-fusion conformation F. All of these studies contribute to an understanding of the diversity of residues on the surface of F that are recognized by the human immune response.

The strengths of the paper are that many antibodies are described, the group went to a lot of effort to perform epitope mapping, and there are some interesting findings suggested about a difference of immunodominance in RSV versus MPV F sites, and perhaps an effect of enzymatic processing on antigenicity of MPV that is a little unusual. A cryoEM structure of one of the leads is provided, which is a nice touch. A relatively high resolution cryoEM structure of delete antibody is provided, which is interesting. There's relatively little mechanistic information gained by this, but the structure is interesting and is an advantage of this manuscript.

The principal glaring weakness of this paper is that there is no in vivo data. There are very straightforward biosafety level 2 animal models for MPV, in fact, several models, for this virus, and other investigators have demonstrated the activity of antibodies in these models. It's hard to interpret the antiviral activity of monoclonal antibodies or their promise for prevention or therapy without in vivo studies.

Specific comments.

Line 163, "knockout" is not really a proper term, in fact I'm not sure what is being done here. Is this an alanine scanning library? The technique should be clarified, and the methods provided in the paper.

Line 200, curious that antibodies were not discovered binding to the equivalent of the RSV site zero. Potentially this is due to the sorting anagen, alternatively the immunodominance pattern that the authors prefer. But they should point out that there may be technical limitations in the way that they isolated antibodies that could have led them to miss antibodies to this site.

Line 227, 'binders' is a colloquial term.

Line 284, in several places in this paper the investigators emphasize that they have found four new antigenic sites or novel sites or new sites. For the most part, the epitopes are peripheral to existing known sites. Emphasizing novelty and new comes across as unnecessary overstatement here.

Line 289, the authors put forth a very provocative statement that there is a trimer-interface-recognizing antibody in the paper, but then provide no other data or description and allude to a manuscript in preparation that is not provided in supplemental information. It's hard to interpret this text.

The full Fv nucleotide sequences of all the antibodies should be deposited in the GenBank, so that other investigators can replicate this work for reproducibility.

Table 1, gene segment designations should be indicated in italics

The "knockout methods" are not provided.

Reviewer #2:

Remarks to the Author:

The report by Xiao et al., describes the profiling of antibody response against human metapneumovirus (hMPV). B cells from patients were sorted and screened for anti-F protein activity using a pre-fusion F trimer that was not proteolytically processed, but otherwise retains the structure of pre-fusion F that is processed (based on previously reported structural, biochemical studies). A number of antibodies were identified and the epitopes targeted by the panel of antibodies was assessed primarily by variety of methods including ELISA, competition biolayer interferometry, cryo-EM, and mutation of putative epitope residues. While previous studies have described antibodies against hMPV, the present study provides a broad survey of the epitopes that are targeted by IgG elicited by natural infection. As such, it provides a useful framework for understanding the immune response against hMPV.

While in sum the broad stroke identification of epitopes targeted by diverse mAbs is valuable, the way these antibodies were analyzed (one by cryo-EM and hydrogen deuterium exchange-MS, others by ELISA with a relatively sparse sampling of "knock out mutations", some by selection for escape variants with propagation in the presence of antibody, some by BLI competition... muddles the overall picture and makes comparison among antibodies somewhat difficult to appreciate. In part some of this relates to how and which parts of data are discussed in the text; there is sometimes more data that is presented in figures but only briefly discussed. This may be a limitation due to the impressive amount of data that is included in this study. But in some cases the choice of things to focus upon is a bit puzzling. For example the potent antibody M4B06 that is selective for pre-fusion F was examined by cryo-EM and HDX-MS, but these data and choice of antibody-antigen pairing, while interesting, offer little to help analyze the other areas that were discussed in much greater detail relating to selectivity for processed vs unprocessed pre-F immune responses.

The depletion assay indicated that antibody titers specific for processed vs unprocessed preF were largely similar (with unprocessed pre-F having the most substantial depletion, indicating promiscuity of this antigen's interaction with antibodies). Surprisingly post-F also significantly reduced antibody titers even for antibodies that recognize pre-F forms. This result seems contrary to what we normally anticipate with antibodies that preferentially recognize pre-fusion forms of fusion proteins being active due to their ability to prevent the fusion related conformational changes in F and analogous proteins. The authors only briefly discuss this around line 300. More discussion is warranted given the contrast to what has been demonstrated for other related viruses such as RSV.

Since BLI is used as the basis for much of the quantitative comparisons that were made, I would strongly recommend the BLI measurements be performed with non-specific binding accounted for by

the standard double reference approach. Often dissociation curves appear flat or even slope upward but this is an artifact due to low levels of nonspecific analyte interaction with the sensor that are not being corrected for by a basic buffer reference subtraction. Without the dissociation rates to obtain a KD, using the response magnitude alone is somewhat dicey for quantitative comparisons due to numerous factors that can influence that metric.

An approach such as epitope mapping by negative stain electron microscopy could help tie the observations together and provide much better structural context to interpret the results gained by other approaches. Such data would be highly relevant to the antibody survey as presented.

Reviewer #3:

Remarks to the Author:

This manuscript describes the isolation and in depth characterization of human antibodies targeting the human metapneumovirus fusion protein. Overall, the manuscript is quite exciting and provides several new findings for the field, including the identification of hMPV F-specific mAbs targeting new epitopes. The study used very thorough techniques to fully characterize the mAbs. The manuscript can be accepted after a minor revision detailed below.

Line 19: belongs to "the" Pneumoviridae family

Line 32: Remove "the" in: due to the additional glycosylation

Line 52: "The" F protein

Line 86: Include references to:

- a. Human antibody recognition of antigenic site IV on Pneumovirus fusion proteins, 2018
- b. Structural basis for antibody cross-neutralization of respiratory syncytial virus and human metapneumovirus, 2017
- c. Cross-neutralization of four paramyxoviruses by a human monoclonal antibody, 2013
- d. A Broadly Neutralizing Human Monoclonal Antibody Exhibits In Vivo Efficacy Against Both Human Metapneumovirus and Respiratory Syncytial Virus, 2015

Line 130: Among "the" isolated mAbs

Line 144 and extended data Figure 4, provide EC50 values for the control antibody.

Line 160: reference site V papers

- a. Rapid profiling of RSV antibody repertoires from the memory B cells of naturally infected adult donors
- b. A novel pre-fusion conformation-specific neutralizing epitope on the respiratory syncytial virus fusion protein

Line 181: For the first group of mAbs, Bar-Peled et al previously reported site III mAbs that do not cross neutralize RSV F, and do not compete with DS7. Please clarify how this group compared to MPV364 in the results and discussion. This paper describes site IIIa.

Some data needs to be provided for the M2E04-like internal epitope. How was the internal epitope determined? Was there preferential binding to monomeric hMPVF? References to this describe manuscript in preparation, but there is no clear information on how it was discovered that this group of mAbs binds an internal epitope.

Figure 2. Although the structure of the site V antibody was determined to 3.75 Angstroms, additional figures of the interacting regions are needed. It should be possible to at least determine the secondary structures mediating the interaction between F and Fab and these should be displayed in the Figure. Why not at least build the variable region structure into the map to fully define the epitope? Is the epitope for this antibody contained within 1 protomer or does it cross protomers? Are there glycans near the epitope?

Make the format of "IC50" consistent.

For Fig 4b, please also include a map of the residues of epitopes in the sequence of hMPV F, the circles in the current figure look too general

Do the site III' and IV' epitopes span two protomers sites?

For the processed/unprocessed preF used in the Differential Scanning Fluorimetry and Serum absorption assay, are they in monomeric or trimeric form, or a mixture? Please specify in the methods.

How was recombinant protein mutants for each of the epitopes in Figure 3 decided upon?

Line 384: What type of biotinylation was used, random or avi-tagged?

Lines 527-545: Describe the media for hMPV growth, presumably without FBS. How were plaques identified for plaque picking?

Data availability: The cryo-EM map/structure need to be deposited and accession numbers provided. Will the complete antibody sequences be deposited? At a minimum, the sequence of M4B06 from the structure should be provided with the structure deposition.

We would like to thank the editors and all three reviewers for all the suggestions aiming to improve our manuscript. The scope of this study is to characterize a large panel of hMPV F targeting antibodies and understand their epitopes, neutralization activities, and antigen binding specificity (or preference). We have revised the manuscript based on the suggestions to organize the data in a better format and prepared the manuscript with improved clarity. Please find our responses to reviewers' comments below.

REVIEWER COMMENTS

Reviewer #1 (Remarks to the Author):

This is a solid paper that describes an antibody discovery campaign for human metapneumovirus, targeting the fusion protein. This is a logical target, given that previous literature on our RSV and MPV have shown that the F protein is the principal neutralizing target for this virus. A lot of work has been done in this field on RSV, but much less work has been done on MPV. MPV and RSV cross reactive antibodies also have been described. Therefore, this is a mature field, but this paper adds a lot more MPV F antibodies into the literature and thus, could be a substantial contribution. There are a lot of words like novel and 1st in the paper that are a bit of a stretch, and all that language could be removed from this paper, without denigrating the presentation of the data.

The approach is to use single B cell technologies to isolate naturally occurring human monoclonal antibodies from human donors who have been previously infected naturally. The investigators use common techniques of proteomics and escape mutants for epitope mapping to identify epitopes, or at least the major antigenic sites, identified by these antibodies. There is also a lot of work done on the recognition of pre- versus post-fusion conformation F. All of these studies contribute to an understanding of the diversity of residues on the surface of F that are recognized by the human immune response.

The strengths of the paper are that many antibodies are described, the group went to a lot of effort to perform epitope mapping, and there are some interesting findings suggested about a difference of immunodominance in RSV versus MPV F sites, and perhaps an effect of enzymatic processing on antigenicity of MPV that is a little unusual. A cryoEM structure of one of the leads is provided, which is a nice touch. A relatively high resolution cryoEM structure of delete antibody is provided, which is interesting. There's relatively little mechanistic information gained by this, but the structure is interesting and is an advantage of this manuscript.

The principal glaring weakness of this paper is that there is no in vivo data. There are very straightforward biosafety level 2 animal models for MPV, in fact, several models, for this virus, and other investigators have demonstrated the activity of antibodies in these models. It's hard to interpret the antiviral activity of monoclonal antibodies or their promise for prevention or therapy without in vivo studies.

Response: We would like to thank reviewer #1 for the constructive and positive review. We have revised the manuscript throughout based on the comment provided to avoid overstatement on novelty. The primary focus of this manuscript is to characterize a large panel of MPV antibodies from human memory B cells. While we agree with reviewer #1 that in vivo study would be one step further to validate the antiviral activity of monoclonal antibodies, we think such study is out of scope of

current manuscript. Although there are exceptions that mAbs with weak or no *in vitro* neutralizing activities show protection in animal models, mAbs with potent *in vitro* neutralizing activities on RSV and MPV have been shown to extend to *in vivo* protection, including many RSV neutralizing mAbs, as well as MPV antibodies 54G10 and DS7^{1,2}. Therefore, we do not anticipate any substantial difference from what has been reported in the literature for the antibodies described in this manuscript, with respect to *in vitro* neutralization potency and corresponding *in vivo* protection. We are currently conducting *in vivo* efficacy studies on selected Abs described here, and such data will be subject of future publications.

Line 163, “knockout” is not really a proper term, in fact I'm not sure what is being done here. Is this an alanine scanning library? The technique should be clarified, and the methods provided in the paper.

Response: We understand the reviewer’s concern calling the mutational changes at antigenic sites defined by antibody binding as “knockout”. We did not perform comprehensive alanine scanning mutagenesis. Instead, we rationally tested a lot of mutants with different designing strategies, such as mutations in known RSV and hMPV F epitopes based on antibody binding and F antigen structures, sequence differences of F proteins between hMPV A and B viruses, or epitopes determined by other experimental approaches. Some of them did not work (for example with very low expression levels), thus we only included the representative mutants that expressed well in this study. Specifically, the mutants reported in this study are derived from several strategies: 1) Based on reported structures of RSV antibodies, we selected residues critical for RSV antibody binding, and mutated the equivalent residues and surrounding residues on hMPV F, assuming the binding mode of hMPV antibodies are analogous to the RSV antibodies. We used this approach to design the site II mutations A238R and R156A/S232A/A238R, and site IV mutation D414K; 2) Based on previously reported RSV epitope mutations³, we mutated equivalent residues on hMPV F and identified site V/AM14-like epitope mutations L130S/G152K/N153A; 3) Some antibodies only neutralize hMPV B but not hMPV A, so we mutated the hMPV F B-strain specific residues to A-strain specific residues to map these antibodies, including site V mutations T143K, N135T/G139N/Y233N, and site IV’ mutations D296K; 4) M4B06 epitope mutations were designed based on the MARM sequences. We have revised the “Epitope mutant design and ELISA assay” (line 603-614) in the method section.

Line 200, curious that antibodies were not discovered binding to the equivalent of the RSV site zero. Potentially this is due to the sorting anagen, alternatively the immunodominance pattern that the authors prefer. But they should point out that there may be technical limitations in the way that they isolated antibodies that could have led them to miss antibodies to this site.

Response: thank you for the suggestion, we have included the potential technical limitation in the discussion section (line 325-328) of revised manuscript.

Line 227, ‘binders’ is a colloquial term.

Response: We have changed “Pre/Post-dual binders” into “Pre/Post-dual binding mAbs”.

Line 284, in several places in this paper the investigators emphasize that they have found four new antigenic sites or novel sites or new sites. For the most part, the epitopes are peripheral to existing known sites. Emphasizing novelty and new comes across as unnecessary overstatement here.

Response: As indicated, we have revised the manuscript to avoid overstatement on novelty and removed words like “novel” and “first”.

Line 289, the authors put forth a very provocative statement that there is a trimer-interface-recognizing antibody in the paper, but then provide no other data or description and allude to a manuscript in preparation that is not provided in supplemental information. It's hard to interpret this text.

Response: We have attached the manuscript that reports the structure of M8C10, a trimer-interface-recognizing antibody for your reference. We plan to submit this manuscript for publication soon.

The full Fv nucleotide sequences of all the antibodies should be deposited in the GenBank, so that other investigators can replicate this work for reproducibility.

Response: Thank you for your suggestion. The sequences reported in this manuscript have been already filed in a provisional patent and will be disclosed soon. In the interim, for any request of these antibodies and related sequences before the patent disclosure, we can provide sequence or antibodies under a material transfer agreement.

Table 1, gene segment designations should be indicated in italics

Response: We have changed the germline designations in Table 1 and Extended Data Table 1 in italics.

The “knockout methods” are not provided.

Response: We have revised the “Epitope mutant design and ELISA assay” (line 603-614) in the methods section.

Reviewer #2 (Remarks to the Author):

The report by Xiao et al., describes the profiling of antibody response against human metapneumovirus (hMPV). B cells from patients were sorted and screened for anti-F protein activity using a pre-fusion F trimer that was not proteolytically processed, but otherwise retains the structure of pre-fusion F that is processed (based on previously reported structural, biochemical studies). A number of antibodies were identified and the epitopes targeted by the panel of antibodies was assessed primarily by variety of methods including ELISA, competition biolayer interferometry, cryo-EM, and mutation of putative epitope residues. While previous studies have described antibodies against hMPV, the present study provides a broad survey of the epitopes that are targeted by IgG elicited by natural infection. As such, it provides a useful framework for understanding the immune response against hMPV.

While in sum the broad stroke identification of epitopes targeted by diverse mAbs is valuable, the way these antibodies were analyzed (one by cryo-EM and hydrogen deuterium exchange-MS, others by ELISA with a relatively sparse sampling of “knock out mutations”, some by selection for escape variants with propagation in the presence of antibody, some by BLI competition... muddles the overall picture and makes comparison among antibodies somewhat difficult to appreciate. In part some of this relates to how and which parts of data are discussed in the text; there is sometimes more data that is presented in figures but only briefly discussed. This may be a limitation due to the impressive amount of data that is

included in this study. But in some cases the choice of things to focus upon is a bit puzzling. For example the potent antibody M4B06 that is selective for prefusion F was examined by cryo-EM and HDX-MS, but these data and choice of antibody-antigen pairing, while interesting, offer little to help analyze the other areas that were discussed in much greater detail relating to selectivity for processed vs unprocessed pre-F immune responses.

Response: We would like to thank reviewer #2 for all the suggestions and concerns. Due to competing resources for the individual methods to delineate the binding epitopes at our institute, we had to rely on all the methods that could help identify the epitope. In our hands, all these methods reliably predict the binding epitopes and therefore the final identified epitope or key residues of the epitope is independent of the method employed. We have revised our manuscript to indicate that using multiple methods, we were able to identify the epitopes more quickly than using a universal method for all antibodies. The purpose of this study is to profile 100+ isolated hMPV F specific antibodies and as reviewer #2 mentioned, provide a broad survey of the epitopes that are targeted by IgG elicited by natural infection, and a useful framework for understanding the immune response against hMPV. We rationally tested a lot of mutants with different designing strategies and have revised the “Epitope mutant design and ELISA assay” (line 603-614) in the method section, as indicated above in the response to Reviewer #1. Confirming the binding/no binding of selected antibodies with such epitope mutants provided us the representative antibodies used as reference in epitope binning experiments, and also further validated the results of epitope binning. Since we have isolated large number of antibodies, by leveraging the available resources, we selected M4B06 as an example to further characterize an antibody with ultra-high neutralization potency in detail. M4B06 was also an antibody that we weren’t able to map its epitope by the F epitope mutants. In the revised manuscript, we changed the organizations of paragraphs, to improve the clarity of the manuscript accordingly.

The depletion assay indicated that antibody titers specific for processed vs unprocessed preF were largely similar (with unprocessed pre-F having the most substantial depletion, indicating promiscuity of this antigen’s interaction with antibodies). Surprisingly post-F also significantly reduced antibody titers even for antibodies that recognize pre-F forms. This result seems contrary to what we normally anticipate with antibodies that preferentially recognize prefusion forms of fusion proteins being active due to their ability to prevent the fusion related conformational changes in F and analogous proteins. The authors only briefly discuss this around line 300. More discussion is warranted given the contrast to what has been demonstrated for other related viruses such as RSV.

Response: The mAbs recognizing epitopes that are presented on both PreF and PostF can still block the structural rearrangement from PreF to PostF, potentially by structural hindrance that prevent the PreF to PostF structural rearrangement outside of the epitope. For example, motavizumab and RB1 are neutralizing mAbs that target RSV site II and site IV respectively^{4,5}, and both epitopes are presented on PreF and PostF. We have included more discussion in the discussion section of revised manuscript (line 330-333). Furthermore, there might be a slight possibility that the PostF we used in the serum depletion assay might contain a small fraction of PreF or intermediate forms of F protein which could also deplete PreF specific antibodies.

Since BLI is used as the basis for much of the quantitative comparisons that were made, I would strongly recommend the BLI measurements be performed with non-specific binding accounted for by the standard double reference approach. Often dissociation curves appear flat or even slope upward but this is an artifact due to low levels of nonspecific analyte interaction with the sensor that are not being

corrected for by a basic buffer reference subtraction. Without the dissociation rates to obtain a KD, using the response magnitude alone is somewhat dicey for quantitative comparisons due to numerous factors that can influence that metric.

Response: Thank you for this great suggestion. We agree that having only association constant (or BLI response in a given period of time) measured may not enable the quantitative comparison of binding kinetics between these antibodies. During revising this manuscript, we have selected a panel of representative mAbs for different epitopes and tested them by BLI with the standard double reference approach as you suggested with unprocessed PreF, processed PreF, and postF (as shown below, processed data after subtraction with the reference wells). However, after subtraction with the double reference, the dissociation to the unprocessed PreF and postF remains positive, the dissociation to the processed PreF is negative but very slow, making the accurate Kd calculation difficult. We think this is likely due to the tested antibodies (IgG) are bivalent, and antigens are trimer, so there is a strong avidity effect on BLI biosensor tips, especially for the unprocessed PreF which three protomers may be in an 'open' conformation. Similar observation with unprocessed PreF has also been reported in other studies⁶. Using monovalent Fabs in this experiment might enable more accurate evaluation of the binding kinetics of isolated antibodies. However, even without accurate Kd measurement, we think comparison of association alone (as currently presented in the manuscript) is still valuable to elucidate the binding preferences of antibodies to the different format of hMPV F antigens valuable for better understanding of immune responses against hMPV, as these antigens have different structural conformations, flexibility, and accessibility to different epitopes. We have revised the manuscript to clarify our findings and also indicated the limitations (line 228-232).

An approach such as epitope mapping by negative stain electron microscopy could help tie the observations together and provide much better structural context to interpret the results gained by other approaches. Such data would be highly relevant to the antibody survey as presented.

Response: In this study, we aimed to utilize multiple approaches to profile the antigenic sites of a large panel of antibodies. For epitope binning, we selected a number of reference antibodies, most of which have been mapped to defined epitopes. Together with other approaches reported in this study, we have confidence and sufficient evidence to demonstrate the distinct epitopes on hMPV F antigen. On the other hand, low-resolution negative staining EM data for a limited set of antibodies may still not sufficiently demonstrate that other antibodies grouped into the same epitope bins will bind to the exact same epitope in the same orientation. Therefore, the additional information we could learn from EM negative staining might as well be limited.

We agree with the reviewer that structural information will provide further and better understanding of these epitopes and the mechanisms of neutralization by these antibodies. The team is currently working on determining the high-resolution structures of representative mAbs but we believe that it is out of scope of the current manuscript, which primarily focuses on grouping these isolated mAbs to distinct epitope bins (antigenic sites) and establish the correlation between these sites versus their functions (binding specificity/neutralization). These additional high resolution structural data will be subject of subsequent publications.

Reviewer #3 (Remarks to the Author):

This manuscript describes the isolation and in depth characterization of human antibodies targeting the human metapneumovirus fusion protein. Overall, the manuscript is quite exciting and provides several new findings for the field, including the identification of hMPV F-specific mAbs targeting new epitopes. The study used very thorough techniques to fully characterize the mAbs. The manuscript can be accepted after a minor revision detailed below.

Line 19: belongs to “the” Pneumoviridae family

Line 32: Remove “the” in: due to the additional glycosylation

Line 52: “The” F protein

Line 86: Include references to:

- a. Human antibody recognition of antigenic site IV on Pneumovirus fusion proteins, 2018
- b. Structural basis for antibody cross-neutralization of respiratory syncytial virus and human metapneumovirus, 2017
- c. Cross-neutralization of four paramyxoviruses by a human monoclonal antibody, 2013
- d. A Broadly Neutralizing Human Monoclonal Antibody Exhibits In Vivo Efficacy Against Both Human Metapneumovirus and Respiratory Syncytial Virus, 2015

Line 130: Among “the” isolated mAbs

Response: Thank you for the suggestions and we have edited the manuscript accordingly.

Line 144 and extended data Figure 4, provide EC50 values for the control antibody.

Response: We have added EC50 values in Extended Data Figure 4 as well as Figure 3B and Extended Data Figure 7.

Line 160: reference site V papers

- a. Rapid profiling of RSV antibody repertoires from the memory B cells of naturally infected adult donors
- b. A novel pre-fusion conformation-specific neutralizing epitope on the respiratory syncytial virus fusion protein

Response: We have added these two references.

Line 181: For the first group of mAbs, Bar-Peled et al previously reported site III mAbs that do not cross neutralize RSV F, and do not compete with DS7. Please clarify how this group compared to MPV364 in the results and discussion. This paper describes site IIIa.

Response: Thank you for the suggestion. The manuscript from Bar-Peled et al previously reported a 'site IIIa' mAb that does not cross neutralize RSV F, does not compete with DS7 (DS-7 site), 101F (site IV), but competes with MPE8 and 25P13 (site III). By comparing the epitope binning profiles between the two studies, we think MPV364 may behave similar to the 'site II' mAbs reported in the current study. However, the differences between assay formats (tandem capturing from Bar-Peled's study vs. sandwich based capturing from our current study), different reference mAbs chosen to be used in the two studies, and relatively low resolution determined by epitope binning, complicated the comparison between mAbs from the current study with MPV364. A side-by-side comparison in the same assay, or determination of high-resolution structures of these mAbs will help to further address this question. We have added a sentence in line 302-307 in the discussion section regarding this.

Some data needs to be provided for the M2E04-like internal epitope. How was the internal epitope determined? Was there preferential binding to monomeric hMPV F? References to this describe manuscript in preparation, but there is no clear information on how it was discovered that this group of mAbs binds an internal epitope.

Response: We have attached a manuscript reporting the crystal structure of M8C10, which targets a deeply buried epitope within the trimerization interface of hMPV F. For M2E04, our current data does not sufficiently support that it targets an internal epitope, therefore we named it as 'M2E04-like' epitope as this group of mAbs showed distinct epitope binning profiles than any other sites.

We agree that more characterization data will be needed to conclude what is the exact epitope/sites for this M2E04-like group of antibodies. For discussion here, we do believe that M2E04 likely targets a partially internal epitope for the following reasons:

- 1) Since one of the antibodies, M2E11L, competed with both M2E04-like antibodies and M8C10 (Figure 4A), we hypothesize that these two groups of antibodies target epitopes that are close to each other and M2E04 also targets an epitope that is adjacent to the M8C10 epitope in the internal trimer interface.
- 2) M2E04 showed less binding to processed hMPV PreF trimer (which forms a more stable trimer) than unprocessed hMPV PreF trimer and monomer, similarly to the M8C10 (internal epitope) group (Extended Data Table 4).
- 3) We have also tried to map its epitope by HDX. The HDX data suggested that the only MPV F peptide that showed moderate level of decreased H/D uptake in the presence of M2E04 is a.a. 59-66. A recent antibody MPV458 has been reported to target an partially internal epitope near the fragment 59-66⁷. Although in our HDX data, such degree of signal is not strong enough to make a clear conclusion (thus we did not include this data in the manuscript), our

observation is consistent with the published data on MPV458. The epitope of MPV458 involves the glycan on N57, which could interfere with HDX signal and might partially explain the weak signal we observed in our HDX experiment.

Figure 2. Although the structure of the site V antibody was determined to 3.75 Angstroms, additional figures of the interacting regions are needed. It should be possible to at least determine the secondary structures mediating the interaction between F and Fab and these should be displayed in the Figure. Why not at least build the variable region structure into the map to fully define the epitope? Is the epitope for this antibody contained within 1 protomer or does it cross protomers? Are there glycans near the epitope?

Response: Thank you for the suggestion. The epitope for the M4B06 antibody is contained within 1 protomer. We believe there are likely glycans near the epitope based on the N-glycosylation sites on hMPV F sequence, but the resolution of current CryoEM map is not sufficient to clearly observe the glycans. Our current CryoEM map is not at the desired quality to build in the Fab molecule, as we observed “orientation preference” in the data set where many particles are observed in one orientation while much less in another. A larger amount of data needs to be collected to compensate this effect and enable structure building. As the main purpose of building the cryo-EM map in this study is to confirm the epitope regions of M4B06 mapped by other approaches, we believe that, additionally building the Fab structure into the CryoEM map does not add any new information on the epitope as it is present on the hMPV F molecule.

Make the format of “IC50” consistent.

Response: We have edited all “IC₅₀” to “IC50”.

For Fig 4b, please also include a map of the residues of epitopes in the sequence of hMPV F, the circles in the current figure look too general

Response: We would love to include a map to show exact epitopes in the single-residue resolutions. However, based on the current epitope binning data, the resolution is too low. We assume site II, III, IV, and V are similar to the definition of RSV F. For site III' and IV', further study to determine structures of antibody-antigen complexes will enable us to show the epitopes in high resolution.

Do the site III' and IV' epitopes span two protomers sites?

Response: Currently we don't have evidence to suggest site III' and IV' epitopes span two protomers.

For the processed/unprocessed preF used in the Differential Scanning Fluorimetry and Serum absorption assay, are they in monomeric or trimeric form, or a mixture? Please specify in the methods.

Response: Both processed/unprocessed PreF were trimeric form with GCN4 domain. We have revised the methods and title of Table 2.

How was recombinant protein mutants for each of the epitopes in Figure 3 decided upon?

Response:

Please see our response to Reviewer #1 above. We have revised the “Epitope mutant design and ELISA assay” (line 603-614) in the method section accordingly.

Line 384: What type of biotinylation was used, random or avi-tagged?

Response: The antigen was randomly biotinylated.

Lines 527-545: Describe the media for hMPV growth, presumably without FBS. How were plaques identified for plaque picking?

Response: The growth media does not have FBS. The plaques were identified based on the CPE. We have revised the method section accordingly.

Data availability: The cryo-EM map/structure need to be deposited and accession numbers provided. Will the complete antibody sequences be deposited? At a minimum, the sequence of M4B06 from the structure should be provided with the structure deposition.

Response: Thank you for your suggestion. The sequences reported in this manuscript have been already filed in a provisional patent and will be disclosed soon. In the interim, for any request of these antibodies and related sequences before the patent disclosure, we can provide sequences or antibodies under a material transfer agreement. As responded above, the current cryo-EM map was built without actual M4B06 Fab in it, as our purpose is to confirm the epitope of M4B06 mapped by other approaches.

Reference:

1. Williams, J.V. *et al.* A recombinant human monoclonal antibody to human metapneumovirus fusion protein that neutralizes virus in vitro and is effective therapeutically in vivo. *J Virol* **81**, 8315-8324 (2007).
2. Schuster, J.E. *et al.* A broadly neutralizing human monoclonal antibody exhibits in vivo efficacy against both human metapneumovirus and respiratory syncytial virus. *J Infect Dis* **211**, 216-225 (2015).
3. Gilman, M.S. *et al.* Rapid profiling of RSV antibody repertoires from the memory B cells of naturally infected adult donors. *Sci Immunol* **1** (2016).
4. McLellan, J.S. *et al.* Structural basis of respiratory syncytial virus neutralization by motavizumab. *Nat Struct Mol Biol* **17**, 248-250 (2010).
5. Tang, A. *et al.* A potent broadly neutralizing human RSV antibody targets conserved site IV of the fusion glycoprotein. *Nat Commun* **10**, 4153 (2019).
6. Bar-Peled, Y. *et al.* A potent neutralizing site III-specific human antibody neutralizes human metapneumovirus in vivo. *J Virol* (2019).
7. Huang, J., Diaz, D. & Mousa, J.J. Antibody recognition of the Pneumovirus fusion protein trimer interface. *PLoS Pathog* **16**, e1008942 (2020).

Reviewers' Comments:

Reviewer #2:

Remarks to the Author:

The authors have addressed most of my concerns through edits to the text including adding some discussion of limitations/caveats and by attempting follow-up antibody binding experiments. The study will be a useful resource for those in the field by setting out the landscape of hMPV antigenicity.

Reviewer #3:

Remarks to the Author:

The authors addressed most of the critiques adequately. Please deposit the cryoEM map into EMDB upon publication, and make antibody sequences available as well once patent is filed with publication. Both of these items will be important for the field and would be expected with a publication in Nature communications.

We would like to thank the editors and the reviewers again for all the suggestions aiming to improve our manuscript. We have revised the manuscript based on the suggestions, please find our responses to reviewers' comments below.

REVIEWER COMMENTS

Reviewer #2 (Remarks to the Author):

The authors have addressed most of my concerns through edits to the text including adding some discussion of limitations/caveats and by attempting follow-up antibody binding experiments. The study will be a useful resource for those in the field by setting out the landscape of hMPV antigenicity.

Response: Thank you very much for carefully reviewing our manuscript again.

Reviewer #3 (Remarks to the Author):

The authors addressed most of the critiques adequately. Please deposit the cryoEM map into EMDB upon publication, and make antibody sequences available as well once patent is filed with publication. Both of these items will be important for the field and would be expected with a publication in Nature communications.

Response: Thank you very much for carefully reviewing our manuscript again. We have deposited the CryoEM map into EMDB database (EMD-25810) and updated this information in the "Data Availability" section. We have also added a sentence to address the antibody sequence availability in the "Data Availability" section: "Antibody sequences reported in this manuscript have been filed in a patent application and will be disclosed when it publishes. In the interim, for any request of these sequences before the patent disclosure, we can provide the sequences under a material transfer agreement."